# DiffLiG: Diffusion-enhanced Liquid Graph with Attention Propagation for Grid-to-Station Precipitation Correction

**Yuxiang Li**[1], **Yang Zhang**[1], **Guowen Li**[1], **Mengxuan Chen**[2], **Meng Jin**[4], **Fang Wang**[5],
**Haohuan Fu**[2,3], **Juepeng Zheng**[1,3,†*]

[1]Sun Yat-sen University    [2]Tsinghua University
[3]National Supercomputing Center in Shenzhen    [4]Huawei Technologies Co., Ltd
[5]CMA Earth System Modeling and Prediction Centre

## Abstract

Modern precipitation forecasting systems, including reanalysis datasets, numerical models, and AI-based approaches, typically produce coarse-resolution gridded outputs. The process of converting these outputs to station-level predictions often introduces substantial spatial biases relative to station-level observations, especially in complex terrains or under extreme conditions. These biases stem from two core challenges: (i) **station-level heterogeneity**, with site-specific temporal and spatial dynamics; and (ii) **oversmoothing**, which blurs fine-scale variability in graph-based models. To address these issues, we propose **DiffLiG** (Diffusion-enhanced Liquid Graph with Attention Propagation), a graph neural network designed for precise spatial correction from gridded forecasts to station observations. DiffLiG integrates a GeoLiquidNet that adapts temporal encoding via site-aware OU dynamics, a graph neural network with a dynamic edge modulator that learns spatially adaptive connectivity, and a Probabilistic Diffusion Selector that generates and refines ensemble forecasts to mitigate oversmoothing. Experiments across multiple datasets show that DiffLiG consistently outperforms other methods, delivering more accurate and robust corrections across diverse geographic and climatic settings. Moreover, it achieves notable gains on other key meteorological variables, underscoring its generalizability and practical utility.

## 1 Introduction

**Weather forecasting** has gained increasing attention in both research and society, driven by rapid advances in forecast accuracy and temporal resolution. On one hand, traditional Numerical Weather Prediction (NWP) systems, such as the European Centre for Medium-Range Weather Forecasts (ECMWF)'s Integrated Forecasting System - High Resolution (IFS-HRES) and Integrated Forecasting System - Ensemble (IFS-ENS), have achieved remarkable success in multiscale forecasts through high-resolution physical modeling, data assimilation, and ensemble strategies. On the other hand, the recent surge in AI-based weather models has introduced a data-driven paradigm that bypasses explicit physical parameterizations [27, 32, 34], offering superior scalability and efficiency. For instance, FourCastNet [30] leverages Fourier neural operators for spatio-temporal coupling, Pangu-Weather [5] employs 3D attention to improve mid-range accuracy, and GraphCast [22] utilizes spherical GNNs for 10-day forecasts. FuXi [10] demonstrates a strong potential to extend the forecast horizons through a hierarchical transformer framework tailored for long-range weather prediction.

---

[*†]Corresponding author: `zhengjp8@mail.sysu.edu.cn`

39th Conference on Neural Information Processing Systems (NeurIPS 2025).

Despite these advances, both the NWP and AI models primarily operate on regular-resolution grid outputs, which are insufficient for real-world applications requiring high-resolution, station-level forecasts. In practice, operational users (such as urban planners, airport operators, or hydrological managers) require precise local forecasts at specific stations. However, model outputs often fail to generalize across diverse topographies or under sparse observational coverage, leading to spatial biases and degraded local accuracy. We provide a detailed empirical analysis of such interpolation errors in the Appendix C.

**A further concern arises from the evaluation process itself**: most AI-based forecasting models are trained and validated on reanalysis data such as ERA5 [17], which integrates historical observations with simulated fields. As illustrated in Figure 6, this integration can introduce systematic biases, particularly in regions with sparse observations or complex terrain. While temporally consistent and spatially smooth, these products are not equivalent to real-world station observations. As noted by Ramavajjala & Mitra [33], models like FourCastNet may outperform NWP systems in ERA5, but fail to retain such advantages when evaluated in ground-truth datasets (*e.g.*, NOAA's MADIS), exposing a considerable gap between simulation data and realism data in current AI weather systems.

Therefore, to enable accurate and reliable station-level forecasting, it is crucial to move beyond uniform, coarse-to-fine interpolation strategies and develop models that can adapt to the heterogeneous nature of stations while preserving local spatial structures. This calls for a dedicated interpolation framework that is both station-aware and uncertainty-informed, capable of correcting the structural mismatch between gridded forecasts and real-world observations.

Station-level interpolation faces two fundamental challenges: **station heterogeneity** and **oversmoothing**. The former refers to the diverse geographic, climatic, and observational characteristics across stations. However, most existing interpolation methods adopt uniform architectures or fixed graph structures, without explicitly adapting to the localized properties of individual stations, which often leads to suboptimal generalization and increased local errors, particularly in regions with complex terrain or sparse observations. Oversmoothing in interpolation is another widely observed issue, where predicted values lack spatial variability and fail to preserve local extremes, limiting their usefulness in tasks such as extreme event detection or fine-grained forecast correction. To better understand the origin of these challenges, we conduct a detailed empirical analysis in Appendix D. To address these issues, we propose **DiffLiG** (*Diffusion-enhanced Liquid Graph with Attention Propagation*), a graph neural network framework specifically designed to model station heterogeneity and alleviate oversmoothing in station-level interpolation.

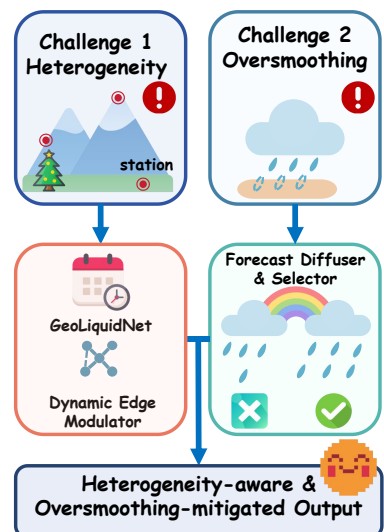

Figure 1: **Two core challenges in station-level interpolation. Heterogeneity** across stations is addressed by the GeoLiquidNet, which models site-specific temporal dynamics, and the Dynamic Edge Modulator, which learns spatially adaptive graph structures. **Oversmoothing** is mitigated by the Probabilistic Diffusion Selector, which generates and selects diverse correction candidates to preserve structural variability.

To overcome these challenges, DiffLiG introduces a unified, uncertainty-aware interpolation framework composed of three key components: a GeoLiquidNet, which combines temporal attention over gridded inputs with a site-aware liquid neural network to capture station-specific temporal dynamics; a Dynamic Edge Modulator, which adaptively configures spatial graphs by jointly learning edge existence, directional attention, and distance-aware influence; and a Probabilistic Diffusion Selector, which generates multiple candidate corrections through noise-perturbed graph propagation and selects or aggregates them via a context-aware scoring mechanism, thereby enhancing both structural variability and predictive reliability [29, 23, 13].

In summary, our contributions are as follows:

• We propose **DiffLiG**, a novel graph neural network framework designed to bridge the gap between gridded forecasts and station-level observations, targeting station heterogeneity and oversmoothing.

• We design three specialized modules to address station heterogeneity and prediction oversmoothing: a GeoLiquidNet for adaptive temporal encoding based on attention and site-aware OU dynamics; a Dynamic Edge Modulator for configuring spatially adaptive graphs through directional attention and learnable edge selection; and a Probabilistic Diffusion Selector for generating diverse correction candidates and selecting reliable outputs under uncertainty.

• Extensive experiments across multiple forecasting sources, variables, and geographic regions demonstrate that DiffLiG consistently outperforms traditional interpolation and GNN baselines in both accuracy and robustness at the station level.

## 2 Related Work

**Grid-based forecasting.** Grid-structured modeling is the backbone of both numerical and AI-driven weather prediction systems. Operational centers such as ECMWF deploy deterministic high-resolution forecasts (IFS-HRES) and ensemble-based systems (IFS-ENS) by solving physical equations over structured grids. Recent AI models have inherited this gridded formulation: FourCast-Net [30] employs Fourier neural operators for global fields; GraphCast [22] uses spherical GNNs for medium-range forecasts; Pangu-Weather [5] introduces 3D convolutional architectures for volumetric prediction. Other large-scale efforts include FuXi [10] and Fengwu [9], which are trained on global reanalysis datasets and have demonstrated competitive performance in medium-range forecasting tasks. Foundation models such as AtmoRep [24], Aurora [6], and ClimaX [28] further decouple forecasting from supervision by pretraining over multi-variable grid datasets. However, these models produce outputs aligned to fixed grid structures and often struggle to adapt to sparse, topography-sensitive station layouts. This mismatch limits their direct applicability in real-world settings where localized accuracy is critical. Our work bypasses the grid altogether, targeting station-level prediction with geometry-aware spatial adaptation.

**From grids to stations.** Traditional spatial interpolation methods, such as inverse distance weighting (IDW), bilinear interpolation, kriging, and spline interpolation, efficiently generate point forecasts from gridded fields, require no training data, and offer clear interpretability with low computational cost. More recently, learning-based approaches have sought to improve accuracy and capture complex dependencies: Bentsen et al. [44] apply a GNN to station time series for wind prediction, MetNet-3 [2] fuses satellite and station inputs via a transformer-style U-Net (albeit on a regridded 4 km mesh), Zhao et al. [4] further highlight the smoothing bias in ERA5 during extreme tropical cyclone rainfall events and propose a deep learning-based correction framework, and MGNN [39] combines ERA5 and MADIS in a multimodal graph for deterministic forecasts. However, all these methods either depend on grid-aligned inputs or outputs, fail to adapt to station-specific heterogeneity, and lack mechanisms for uncertainty quantification or structural diversity. In contrast, our framework operates natively off-grid, leveraging a geometry-aware graph with adaptive edge modulation and geo-liquid temporal dynamics, together with diffusion-based ensemble generation to preserve fine-scale variability and capture predictive uncertainty.

## 3 Method

### 3.1 Problem Formulation

We address the problem of spatial interpolation from structured gridded meteorological forecasts to irregularly distributed observation stations. Our goal is to develop a lightweight and scalable graph neural network (GNN) framework that ensures physical consistency, strong spatial generalization, and the ability to model predictive uncertainty.

Formally, the model takes as input a triplet $(\mathcal{X}, \mathcal{P}, \mathcal{H})$:

- $\mathcal{X} = \left\{ \mathbf{X}^{(t)} \in \mathbb{R}^{C \times H \times W} \mid t = -T_1, \ldots, 0, \ldots, +T_2 \right\}$ denotes the gridded forecast data over $T = T_1 + T_2 + 1$ time steps. Each $\mathbf{X}^{(t)}$ is a multichannel spatial field (e.g., precipitation, wind) with $C$ variables over a regular grid of size $H \times W$ (e.g., $721 \times 1440$ for $0.25°$ global resolution).
- $\mathcal{P} = \left\{ p_j \in \mathbb{R}^2 \mid j = 1, \ldots, M \right\}$ represents the spatial coordinates (latitude and longitude) of $M$ observation stations.

- $\mathcal{H} = \left\{ h_j^{(t)} \in \mathbb{R} \;\middle|\; j = 1, \dots, M;\; t = -T_1, \dots, 0 \right\}$ contains the historical observations at each station over $T_1 + 1$ time steps (from $t = -T_1$ to $t = 0$).

The objective is to learn a mapping function:

$$\mathcal{F} : (\mathcal{X}, \mathcal{P}, \mathcal{H}) \longrightarrow \hat{\mathcal{Y}}, \quad \text{where} \quad \hat{\mathcal{Y}} = \{\hat{y}_j \in \mathbb{R} \mid j = 1, \dots, M\}$$

that predicts the target variable at each station for the future time step $t = +T_2$. The model is trained to minimize the prediction error with respect to the ground truth:

$$\hat{y}_j \approx y_j^{(+T_2)}, \quad \forall j = 1, \dots, M.$$

## 3.2 DiffLiG Overview

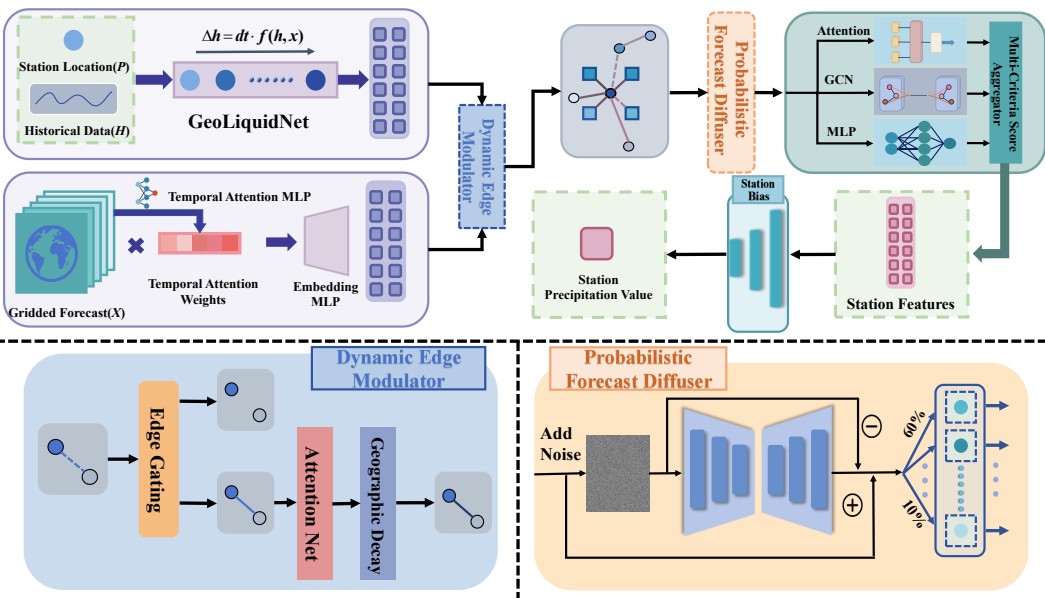

Figure 2: **Overview of DiffLiG.** The model takes gridded meteorological forecasts $\mathcal{X}$, station coordinates $\mathcal{P}$, and historical station observations $\mathcal{H}$ as input. The **GeoLiquidNet** module encodes temporal features for each station based on $\mathcal{H}$ and $\mathcal{P}$. The **Forecast Encoder** assigns temporal attention weights across different time steps of $\mathcal{X}$, followed by an embedding MLP to extract forecast representations. These representations are integrated by the **Dynamic Edge Modulator**, which constructs adaptive graph edges through edge gating, attention net, and geographic decay. The **Probabilistic Forecast Diffuser** generates multiple perturbed predictions to capture epistemic uncertainty. A **Multi-Criteria Forecast Selector** evaluates and selects the most reliable prediction among them. Finally, the selected prediction is refined by the **Station Bias Module**, which incorporates site-specific adjustments through an output MLP, producing the final forecast $\hat{\mathcal{Y}} = \{\hat{y}_j\}_{j=1}^{M}$, where $\hat{y}_j \approx y_j^{(+T_2)}$.

## 3.3 GeoLiquidNet

To handle the non-stationary nature of precipitation sequences, where relevant timescales vary across weather systems[26], regions, and station density, we propose **GeoLiquidOU**, a module that learns spatially-conditioned time constants to adaptively control hidden state transitions. Inspired by Liquid Time-constant Networks (LTC) [15], which model time constants as input-dependent variables to capture nonlinear dynamics, GeoLiquidOU inherits this property to adaptively modulate temporal sensitivity based on each site's historical inputs. Neuroscientific insights further support this design: the diversity of timescales in the brain reflects the dynamic nature of encoded information [41], analogous to varying temporal patterns in meteorological data. GeoLiquidOU thus integrates dynamic temporal encoding with location-aware adaptability, enhancing both interpretability and flexibility in modeling site-specific temporal patterns. Further analytical discussions on the OU-based dynamics are deferred to the Appendix E.

Formally, for a station with input sequence $h = \{h^{(1)}, h^{(2)}, \ldots, h^{(T)}\}$ and location $p \in \mathbb{R}^2$, the module first embeds each timestep via $z^{(t)} = \phi_{\text{proj}}(h^{(t)})$, where $\phi_{\text{proj}}(\cdot)$ is a learnable projection MLP that maps the raw observation $h^{(t)}$ into a latent feature space.

A learnable location encoder $\tau(p)$ then generates the time-step controller as:

$$\tau(p) = \text{Sigmoid}(W_2 \cdot \tanh(W_1 \cdot p)) + 0.5,$$

where $W_1$ and $W_2$ are linear transformation matrices. The output $\tau(p)$ serves as a positive location-dependent scaling factor that defines a site-specific time interval $\Delta t = \tau(p) \cdot \Delta t_{\text{base}}$.

The latent state $\mathbf{z}_t \in \mathbb{R}^d$ evolves through a parameterized Ornstein–Uhlenbeck (OU) process:

$$\Delta \mathbf{z}_t = \Delta t \cdot \left[ \boldsymbol{\gamma} \cdot (\Theta(\mathbf{z}^{(t)}) - \mathbf{z}_t) + \boldsymbol{\sigma} \cdot \boldsymbol{\varepsilon} \right], \tag{1}$$

where $\Theta(\cdot)$ is a learnable MLP that predicts the target equilibrium state, $\boldsymbol{\gamma}, \boldsymbol{\sigma} \in \mathbb{R}^d$ are trainable dynamic coefficients controlling the deterministic and stochastic components, and $\boldsymbol{\varepsilon} \sim \mathcal{N}(0, \mathbf{I})$ denotes Gaussian noise.

The process is initialized with $\mathbf{z}_0 = \phi_{\text{init}}(h^{(1)})$, and iteratively updated over $T$ steps. The final state $\mathbf{z}_T$ thus encodes the historical evolution of each site, capturing temporal dynamics modulated by its geographic location. This continuous-time formulation allows GeoLiquidNet to flexibly adapt to diverse temporal rhythms across stations, enhancing generalization and interpretability in sequence modeling.

### 3.4 Spatial Message Passing Network

To model the spatial heterogeneity and directional propagation of meteorological variables, we introduce a **Spatial Message Passing Network (SMPN)**. It adaptively learns graph structure and edge importance through four components: edge selection, distance-aware decay, directional attention, and residual node update.

**Edge Selection Mechanism.** We employ a neural discriminator $\psi(\cdot)$ to predict the existence probability of an edge based on node coordinates and their Haversine distance: $g_{ij} = \mathcal{G}(\psi([\mathbf{p}_i, \mathbf{p}_j, d_{ij}]))$, where $\mathcal{G}(\cdot)$ denotes a learnable gating function that outputs an edge existence probability. During training, a binary gate $\tilde{g}_{ij} \in \{0, 1\}$ is sampled from $g_{ij}$ to enable data-driven, spatially adaptive edge pruning [12, 19].

**Distance-aware Decay.** To reflect spatial attenuation, each edge is weighted by a decay factor $\kappa_{ij} = \exp(-\beta d_{ij}^2)$, where $d_{ij}$ is the Haversine distance. The decay rate $\beta$ is learned via a bounded sigmoid function, allowing smooth, interpretable modulation of long-range influence.

**Multi-head Attention.** To capture directional dependencies, we design an interaction-aware attention mechanism [37, 40, 8]. For each edge, the input vector $\mathbf{e}_{ij} = [\mathbf{h}_i \| \mathbf{h}_j \| (\mathbf{p}_i - \mathbf{p}_j)]$ is processed by $M$ attention heads. Normalized scores $\alpha_{ij}^{(m)}$ determine the contribution of neighbor $j$ to $i$, enabling asymmetric and context-sensitive aggregation [42].

**Residual Update.** The final propagation weight is $w_{ij} = \tilde{g}_{ij} \cdot \kappa_{ij} \cdot \alpha_{ij}$. Messages $m_{ij}$ are generated via $\phi_{\text{msg}}$ and aggregated with normalization. Each node is updated via a residual MLP [7, 16]:

$$\mathbf{h}_i' = \mathbf{h}_i + \phi_{\text{upd}}([\mathbf{h}_i, \bar{\mathbf{m}}_i])$$

This enables stable, fine-grained updates across two stages: first from gridded fields to stations, then among stations for local refinement.

---
**Algorithm 1: Spatial Message Passing Network**

**Input:** Node features $\mathbf{h}_i$, positions $\mathbf{p}_i$, number of spatial neighbors $k$, edge gates $\tilde{g}_{ij}$, decay factors $\kappa_{ij}$, attentions $\alpha_{ij}$

1. For each node $i$, identify the $k$ nearest neighbors $\mathcal{N}_k(i)$ based on Haversine distance
2. Initialize $\bar{\mathbf{m}}_i \leftarrow \mathbf{0}$, count $\leftarrow 0$
3. For each $j \in \mathcal{N}_k(i)$:
   - If $\tilde{g}_{ij} = 1$, compute propagation weight $w_{ij} = \tilde{g}_{ij} \cdot \kappa_{ij} \cdot \alpha_{ij}$
   - Compute message $m_{ij} = \phi_{\mathrm{msg}}([\mathbf{h}_i, \mathbf{h}_j, \mathbf{p}_i - \mathbf{p}_j])$
   - Accumulate: $\bar{\mathbf{m}}_i \mathrel{+}= w_{ij} \cdot m_{ij}$; increment count
4. Normalize: $\bar{\mathbf{m}}_i \leftarrow \bar{\mathbf{m}}_i / \mathrm{count}$
5. Update node state: $\mathbf{h}'_i = \mathbf{h}_i + \phi_{\mathrm{upd}}([\mathbf{h}_i, \bar{\mathbf{m}}_i])$
---

### 3.5 Probabilistic Forecast Diffuser

To capture the inherent uncertainty of meteorological predictions and generate diverse forecast candidates, we propose a **Probabilistic Forecast Diffuser**. This module injects stochastic perturbations into node features and processes them through a graph-based denoising network, yielding a set of sample-level forecasts for subsequent evaluation and selection. A concise mathematical analysis explaining why the diffusion process enables recovery and amplification of extreme precipitation values is provided in Appendix F.

**Sample Generation via Diffusion.** Given node features $x$ and position encodings $\phi(\mathbf{p})$, we generate $N$ samples by adding Gaussian noise $\varepsilon_i \sim \mathcal{N}(0, \sigma^2 I)$ and passing the perturbed inputs through a graph U-Net denoiser $f_{\mathrm{UNet}}$ [18, 35, 1]. Each sample is constructed via residual correction:

$$s_i = \lambda \cdot x + (1 - \lambda) \cdot (x + \varepsilon_i - f_{\mathrm{UNet}}(x + \varepsilon_i + \phi(\mathbf{p}))) \tag{2}$$

Here, $\lambda \in [0, 1]$ is a blending coefficient that controls the contribution of original versus corrected features. The result is an ensemble of spatially coherent yet diverse forecasts $\{s_1, \ldots, s_N\}$.

**Uncertainty Estimation.** We quantify prediction uncertainty for each node by computing the sample variance:

$$u_j = \frac{1}{N} \sum_{i=1}^{N} (s_{i,j} - \bar{s}_j)^2, \quad \bar{s}_j = \frac{1}{N} \sum_{i=1}^{N} s_{i,j} \tag{3}$$

This serves as a natural metric for identifying regions of disagreement across the ensemble and guiding adaptive selection.

### 3.6 Multi-Criteria Score Selector

To extract the most plausible forecast from the generated ensemble, or combine multiple outputs into a robust final prediction, we introduce the **Multi-Criteria Score Selector**. This module evaluates each candidate using three scoring perspectives: semantic consensus, spatial coherence, and physical plausibility [21, 25].

**1. Semantic Consensus.** A self-attention mechanism is applied across all $N$ samples to estimate their mutual agreement. Each sample is assigned an attention-based consensus score, measuring its alignment with the ensemble majority.

**2. Spatial Coherence.** To enforce geophysical smoothness, we pass each sample through a lightweight graph convolution network that estimates the degree of spatial consistency across neighboring stations. This captures the natural continuity of meteorological variables in space.

**3. Extreme Value Detection.** We implement a lightweight MLP-based scoring head that focuses on detecting physically implausible or statistically extreme predictions. It penalizes samples that contain abnormal spikes, unrealistic magnitudes, or values significantly deviating from historical ranges, thereby reducing their influence during ensemble selection.

**Final Selection.** The above three scores are aggregated using a two-layer MLP to compute a unified quality score $q_i$ for each sample. We apply a softmax over all scores to produce sample weights. The final output is selected via either hard picking (the best-scoring $q_i$) or weighted fusion of the ensemble.

## 4 Experimental

### 4.1 Experimental Setup

To comprehensively evaluate the generalization and robustness of the proposed method under various meteorological models and complex geographical conditions, we design a station-level precipitation interpolation task based on multiple forecast products and ground truth observations.

**Data Sources.** We utilize 24-hour accumulated precipitation forecasts from five representative meteorological models, including both traditional numerical systems and emerging AI-based models. Specifically, the selected inputs cover: ECMWF's high-resolution deterministic forecast (IFS-HRES), ensemble forecast system (IFS-ENS), and reanalysis data (ERA5), as well as models such as FourCast-Net, and GraphCast. As ground-truth reference, we employ in-situ measurements from 2288 surface meteorological stations provided by the China Meteorological Administration. These stations are distributed across diverse terrains and climatic regions, introducing strong spatial heterogeneity. All observational data are resampled to daily resolution and temporally aligned with forecast inputs. The resulting dataset forms a 24-hour accumulated precipitation interpolation benchmark for evaluating spatial reasoning and correction capabilities under complex landscapes.

**Task Definition.** The task is formulated as a grid-to-station spatial mapping problem. Each sample consists of coarse-resolution gridded inputs and historical observations at stations, and the model is required to predict the target variable (e.g., 24-hour precipitation) at all station locations.

**Extended Experiments.** To examine the model's variable adaptability and temporal generalization, we also conduct extended experiments on 2-meter temperature prediction using Pangu-Weather temperature products. Detailed results of this auxiliary experiment are included in the Appendix G. We report additional results in the Appendix H, including a sensitivity analysis under varying forecast horizons and structural parameter settings.

| Category | Methods |
|---|---|
| Traditional Interpolation | Nearest Neighbor |
| | Linear Interpolation |
| | Inverse Distance Weighting |
| | Ordinary Kriging |
| Graph Neural Networks | GCN [19] |
| | GAT [38] |
| | GraphSAGE [14] |
| | KCN [3] |
| Physics-Inspired Architectures | MeshGraphNet [31] |
| | MGNN [39] |
| Spatial & Generative | ViT [11] |
| | Diffusion Networks [18] |

Table 1: Classification of Baseline Models

| Model | P (#) | S (MB) |
|---|---|---|
| DiffLiG (ours) | 49,361 | 0.19 |
| LiGAP | 39,397 | 0.15 |
| GAP | 33,650 | 0.13 |
| Graph-bias | 29,506 | 0.11 |
| Graph | 24,930 | 0.10 |
| GCN | 29,031 | 0.11 |
| GAT | 29,287 | 0.11 |
| GraphSAGE | 33,127 | 0.13 |
| KCN | 30,520 | 0.12 |
| ViT | 51,538 | 0.20 |
| Diffusion | 54,014 | 0.21 |
| MeshGraphNet | 41,863 | 0.16 |
| MGNN | 40,741 | 0.16 |

Table 2: Model Complexity Comparison

**Baseline Models.** To assess model performance, we compare against a broad range of interpolation and learning-based baselines, as shown in Table 1. This diverse benchmark suite allows us to evaluate accuracy, structural consistency, and generalization ability across multiple paradigms. A more detailed description of the baseline methods, including their modeling assumptions and limitations, is provided in the Appendix K.

**Model Complexity.** To further examine architectural efficiency, we report the parameter counts and storage sizes of all compared models in Table 2. Although DiffLiG is not the smallest in scale, it introduces only a moderate increase in parameters while achieving significant gains in accuracy and generalization, reflecting an efficient use of model capacity. Its ablation variants reveal the role of individual modules: LiGAP removes the diffusion-based ensemble component from DiffLiG, GAP further discards the Temporal Module to isolate temporal contributions, Graph-bias omits the Spatial

Message Passing Network, and Graph represents the most reduced configuration without the Station Bias Module.

**Evaluation Metrics.** We employ five quantitative metrics to comprehensively assess model performance: Root Mean Squared Error (RMSE), Anomaly Correlation Coefficient (ACC), and Threat Scores (TS) at 2 mm, 20 mm, and 100 mm thresholds. RMSE measures the overall magnitude deviation between predictions and observations, ACC evaluates spatial correlation and pattern consistency across stations, while TS metrics quantify the model's capability to detect precipitation events of increasing intensity, reflecting its sensitivity to light, moderate, and extreme rainfall. Together, these metrics provide a balanced evaluation of numerical accuracy, spatial coherence, and event-level detection skill.

**Implementation and Training Details.** To ensure fair comparison, all baseline models are trained under identical hardware environments, using the same loss function and optimization settings. Further experimental details, including training strategies, configurations, and runtime statistics, are provided in Appendix I.

## 4.2 Experimental Results

| Method | IFS-HRES | | | ERA5 | | | IFS-ENS | | | FourCastNet | | |
|---|---|---|---|---|---|---|---|---|---|---|---|---|
| | RMSE | ACC | TS@2mm | RMSE | ACC | TS@2mm | RMSE | ACC | TS@2mm | RMSE | ACC | TS@2mm |
| DiffLIG (ours) | **5.0883** | **0.7395** | **0.4621** | **4.8021** | **0.7677** | **0.5121** | **4.7208** | **0.7721** | **0.5286** | **5.1014** | **0.7303** | 0.4937 |
| Nearest | 6.0998 | 0.6511 | 0.4121 | 5.2145 | 0.7265 | 0.4693 | 4.9717 | 0.7468 | 0.4801 | 5.3871 | 0.6795 | 0.4865 |
| Linear | 5.9033 | 0.6667 | 0.4011 | 5.1435 | 0.7313 | 0.4808 | 4.9395 | 0.7504 | 0.4757 | 5.3721 | 0.6948 | 0.4858 |
| OK | 5.7423 | 0.6577 | 0.3931 | 5.1011 | 0.7387 | 0.4891 | 4.8932 | 0.7581 | 0.4434 | 5.3088 | 0.7052 | 0.4811 |
| IDW | 5.8274 | 0.6742 | 0.3995 | 5.1058 | 0.7368 | 0.4558 | 4.9299 | 0.7515 | 0.4716 | 5.3683 | 0.6831 | 0.4837 |
| GCN | 5.4832 | 0.6927 | 0.4139 | 5.0680 | 0.7417 | 0.4385 | 4.8731 | 0.7610 | 0.5052 | 5.2313 | 0.7016 | 0.4970 |
| GAT | 5.5444 | 0.6667 | 0.3202 | 5.0401 | 0.7449 | 0.4388 | 4.9137 | 0.7572 | 0.4987 | 5.2551 | 0.6964 | 0.4921 |
| GraphSAGE | 5.6401 | 0.6495 | 0.3754 | 5.1154 | 0.7243 | 0.4634 | 4.9343 | 0.7423 | 0.4922 | 5.2511 | 0.7001 | 0.4936 |
| KCN | 5.3931 | 0.6962 | 0.4158 | 5.0837 | 0.7468 | 0.4498 | 4.8897 | 0.7572 | 0.4925 | 5.2264 | 0.7122 | 0.4923 |
| ViT | 5.7848 | 0.6338 | 0.3626 | 5.1232 | 0.7121 | 0.4428 | 5.2365 | 0.7034 | 0.4832 | 5.3912 | 0.7080 | 0.4812 |
| Diffusion | 5.8480 | 0.5963 | 0.3342 | 5.1000 | 0.7284 | 0.4537 | 5.1742 | 0.7183 | 0.4952 | 5.3781 | 0.6932 | 0.4921 |
| MeshGraphNet | 5.5322 | 0.6767 | 0.4374 | 5.1728 | 0.7283 | 0.4903 | 5.0132 | 0.7332 | 0.5143 | 5.2619 | 0.6931 | **0.4987** |
| MGNN | 5.4428 | 0.6858 | 0.3640 | 4.9904 | 0.7501 | 0.5000 | 4.8542 | 0.7657 | 0.4357 | 5.1878 | 0.7164 | 0.4423 |

Table 3: **Performance Comparison of Interpolation Methods across Multi-Source Forecast Inputs.** Top-3 values in each metric are highlighted with increasing color intensity. Bold values indicate the best performance, while underlined values represent the second-best results for each metric.

To evaluate the model's capability in extracting reliable station-level information from coarse-resolution gridded forecasts, we conduct comprehensive experiments across four representative forecast sources: IFS-HRES, ERA5, IFS-ENS, and FourCastNet. Results are summarized in Table 3.

**DiffLiG demonstrates consistent superiority across all three metrics.** In terms of TS@2mm, which measures the model's ability to identify precipitation events, DiffLiG achieves the highest scores on IFS-HRES (0.4621), ERA5 (0.5121), and IFS-ENS (0.5286). On FourCastNet, it reaches 0.4937, closely following the best-performing baseline (0.4987), underscoring its strong event detection capability. For RMSE, DiffLiG achieves the lowest error across all four sources, indicating more accurate magnitude reconstruction. In terms of ACC, DiffLiG again attains the highest accuracy on each dataset, reflecting its consistent predictive correctness and classification reliability.

Furthermore, the consistent improvements of DiffLiG over all baselines across multiple evaluation metrics suggest its strong capacity to mitigate station-level **heterogeneity**. Unlike traditional methods that often suffer from degraded performance in regions with complex topography or sparse observations, DiffLiG achieves lower RMSE (more accurate magnitude), higher ACC (stronger spatial correlation with observations), and higher TS@2mm (better event detection skill). These results indicate that the model effectively adapts to diverse spatial characteristics and observational conditions, demonstrating its practical utility for heterogeneous station-level interpolation in real-world settings.

**In addition, DiffLiG exhibits strong cross-source generalization.** It maintains either top or second-best performance across all forecast sources, regardless of the underlying modeling paradigm: be it physically based numerical weather prediction (e.g., IFS-HRES, IFS-ENS) or AI-driven models (e.g., FourCastNet). This cross-domain consistency validates the robustness of its architectural design under varying input distributions and error characteristics.

## 4.3 Evaluation of Oversmoothing Mitigation on IFS-HRES

To evaluate the capability of different models in mitigating spatial oversmoothing, we compute TS at 20mm and 100mm thresholds on the IFS-HRES dataset. As shown in Table 4, DiffLiG achieves the highest TS values across both thresholds, demonstrating its effectiveness in alleviating the oversmoothing problem. LiGAP, which removes the diffusion module from DiffLiG while keeping all other components identical, shows a noticeable degradation in both TS@20mm and TS@100mm, highlighting the effectiveness of the probabilistic diffusion mechanism in mitigating the oversmoothing effect inherent in graph propagation.

| Model | TS@20mm | TS@100mm |
|---|---|---|
| DiffLiG (ours) | **0.3122** | **0.0202** |
| LiGAP | 0.2819 | 0.0147 |
| Linear | 0.2764 | 0.0156 |
| GAT | 0.2986 | 0.0162 |
| GCN | 0.2836 | 0.0131 |
| GraphSAGE | 0.2648 | 0.0157 |
| KCN | 0.2862 | 0.0136 |
| MeshGraphNet | 0.2925 | 0.0144 |
| MGNN | 0.2611 | 0.0153 |

Table 4: **Oversmoothing Mitigation on IFS-HRES.**

## 4.4 Visual Evidence of Oversmoothing Mitigation

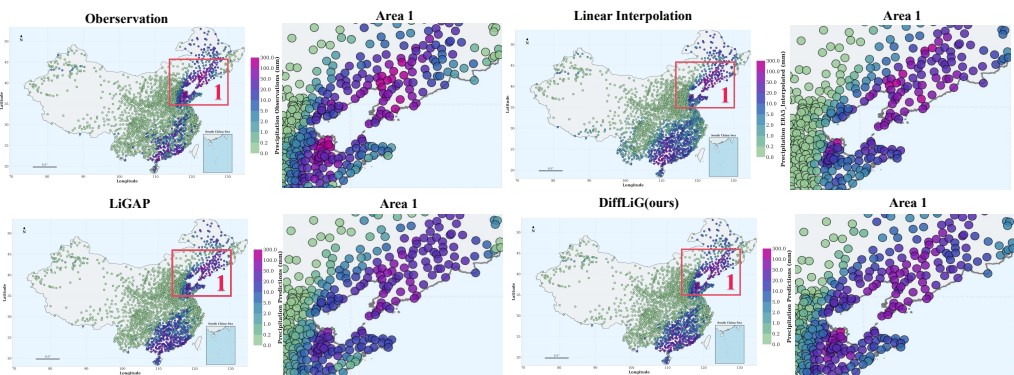

Figure 3: **Visual comparison of spatial interpolation results.** Predicted daily accumulated precipitation for July 5, 2022.

To assess oversmoothing mitigation, we visualize predicted station-level precipitation on July 5, 2022—a day with high spatial variability (Figure 3). Compared to baseline methods, DiffLiG produces sharper and more spatially coherent outputs, capturing localized extremes more effectively. LiGAP, which removes the diffusion module from DiffLiG, shows improved spatial structure but remains conservative due to its deterministic nature. The full DiffLiG model leverages ensemble diversity and selective refinement to better recover fine-scale variability aligned with observations. A more detailed visualization of the results is provided in the Appendix N for further reference.

## 4.5 Cross-Dataset Evaluation on GraphCast for Robustness Validation

| | GraphCast | | |
|---|---|---|---|
| Method | RMSE ↓ | ACC ↑ | TS@2mm ↑ |
| **DiffLiG (ours)** | **5.5172** | **0.7345** | 0.5218 |
| Nearest | 5.7036 | 0.7133 | 0.3520 |
| Linear | 5.6829 | 0.7153 | 0.3515 |
| OK | 5.5923 | 0.7132 | 0.3612 |
| IDW | 5.6772 | 0.7158 | 0.3506 |
| GCN | 5.5517 | 0.7302 | 0.4985 |
| GAT | 5.5773 | 0.7271 | **0.5245** |
| GraphSAGE | 5.8757 | 0.6811 | 0.5001 |
| KCN | 5.5980 | 0.7256 | 0.5181 |
| ViT | 5.8321 | 0.6844 | 0.4212 |
| Diffusion | 5.7342 | 0.6923 | 0.4354 |
| MeshGraphNet | 6.2677 | 0.6246 | 0.5171 |
| MGNN | 5.6537 | 0.7288 | 0.3526 |

Table 5: Performance of different methods on GraphCast.

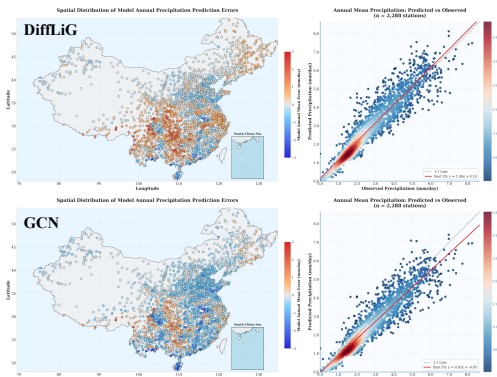

Figure 4: Spatial distribution (left) and scatterplot (right) for DiffLiG vs. GCN.

To evaluate the robustness and generalization capability of our heterogeneity-aware forecasting model, we conduct a cross-dataset inference experiment. Specifically, the model is trained entirely on the **IFS-HRES** dataset and directly applied to the **GraphCast** dataset without any fine-tuning or domain adaptation. The performance comparison is reported in Table 5, and the spatial error distribution alongside station-wise scatter-plot of DiffLiG versus GCN is visualized in Figure 4.

### 4.6 Ablation Studies

**Structural modeling.** Starting from the base model with static edges and uniform aggregation, performance is limited across all metrics. Adding the *Dynamic Edge Modulator* leads to a significant improvement in TS@2mm, highlighting the benefit of learning spatially adaptive connections. The *Station Bias Correction* further improves ACC by capturing site-specific deviations, especially where grid forecasts systematically diverge from observations. These modules enhance the model's ability to reflect spatial heterogeneity, but temporal dynamics remain unmodeled.

**Temporal encoding.** Introducing the *Temporal Module* leads to a substantial improvement in TS@2mm (+16.3%), indicating enhanced sensitivity to localized rainfall events. By jointly encoding gridded sequences and station histories, the model better captures evolving precipitation patterns, leading to concurrent improvements in both ACC and RMSE.

**Uncertainty modeling and selection.** The *Probabilistic Forecast Diffuser* reduces RMSE by modeling ensemble uncertainty, though basic aggregation methods show limited effect on TS and ACC. The final

Figure 5: **Ablation trend.** Module-wise improvements across RMSE, ACC, and TS@2mm.

addition of the *Multi-Criteria Score Selector* brings all metrics to their best levels by prioritizing forecasts with spatial consistency and physical plausibility. This confirms the importance of informed selection in ensemble-based interpolation.

## 5 Conclusion

We presented DiffLiG, a diffusion-enhanced liquid graph framework for high-resolution spatial correction from gridded forecasts to station-level observations. To address the challenge of station heterogeneity, DiffLiG integrates a Geo-Liquid Net that captures site-specific temporal dynamics, and a Dynamic Edge Modulator that independently learns spatially adaptive graph structures tailored to each target station. To mitigate oversmoothing and model predictive uncertainty, a unified Probabilistic Diffusion Selector generates diverse correction candidates via noise-perturbed sampling and refines them through multi-criteria ensemble selection.

Extensive experiments across multiple meteorological datasets and forecast products demonstrate that DiffLiG delivers consistent improvements in both accuracy and event sensitivity over classical interpolation methods and strong graph-based baselines. Its effectiveness extends beyond precipitation, achieving strong performance on other key variables such as temperature, highlighting its flexibility and generalizability.

DiffLiG offers a unified and efficient framework for correcting coarse meteorological forecasts under real-world spatial complexity, with strong potential for application in post-processing, data assimilation, and multi-source fusion tasks. Looking forward, DiffLiG provides a foundation for exploring graph-based spatial correction under more complex scenarios, such as cross-regional generalization, extreme weather adaptation, and integration with multi-modal inputs (e.g., satellite imagery or radar). Future work may also incorporate causal or physical priors to further enhance the interpretability and resilience of correction under evolving climate dynamics.

## Acknowledgements

This work was supported in part by the National Key Researchand Development Plan of China under Grant 2023YFB3002400; in part by the National Natural Science Foundation of China under Grant T2125006 and Grant 42401415; in part by Shenzhen Science and Technology Program under Grant KCXFZ20240903093759004 and Grant KJZD20230923115106012; in part by the Open Research Fund of Pengcheng Laboratory under No.2025KF1B0040; and in part by Jiangsu Innovation Capacity Building Program under Project BM2022028.

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

## Appendices Contents

# A  Broader Impact

This study proposes a lightweight and modular graph-based framework for station-level precipitation correction, designed to enhance the adaptability and spatial resolution of existing weather forecasting systems. Without requiring the retraining of large-scale models, our approach enables flexible post-processing of numerical or AI-generated forecasts, producing locally refined predictions at arbitrary coordinates using only location metadata and limited historical observations. This capability supports fine-grained, adaptive forecasting across diverse geographies, including remote, mountainous, or under-instrumented regions, and has the potential to broaden the accessibility of reliable weather services worldwide.

Our method improves the spatial precision and reliability of extreme precipitation forecasts by explicitly modeling site-specific spatial structure, incorporating dynamic edge modulation, and introducing a diffusion-based ensemble generator with a multi-criteria output selector. This allows more accurate localization of high-risk areas and robust identification of extreme-value signals, thus offering decision-level support for early warning systems related to flooding, torrential rainfall, or hydrological disasters. Unlike coarse-resolution grid-based outputs, the proposed framework allows targeted forecast refinement at critical sites, contributing to improved disaster preparedness and risk mitigation.

In terms of sustainability, the proposed architecture emphasizes post-hoc spatial adaptation rather than end-to-end retraining, reducing the demand for high-performance computing and enabling deployment in low-resource environments. As climate modeling and forecasting scale to global applications, energy-efficient and scalable forecasting solutions are urgently needed. Our model contributes to this trend by providing an extensible correction module that balances accuracy with computational economy.

Beyond meteorology, the modular design and uncertainty-aware structure of our model may benefit other spatiotemporal inference tasks with sparse measurements, such as air quality estimation, ecological monitoring, or environmental hazard detection. The integration of spatial message passing, site heterogeneity handling, and ensemble-based confidence evaluation offers a transferable paradigm for building robust, site-adaptive models across scientific domains.

While we have not identified direct negative societal impacts associated with the proposed method, potential concerns may arise depending on downstream applications. For instance, if the model is deployed without adequate validation in regions with limited ground truth data, it may introduce misleading corrections or overconfident forecasts, especially in high-stakes scenarios. Moreover, localized correction systems, if misused or misinterpreted by non-experts, could inadvertently lead to false alarms or reduced trust in public forecasting services. These risks highlight the importance of responsible deployment and continual calibration against local observations when adapting the model to new domains.

# B  Limitations

While the proposed station-level correction framework demonstrates strong adaptability and predictive performance, several limitations remain that warrant further attention and refinement.

Our current implementation primarily utilizes precipitation and temperature variables from ERA5 as model inputs. Other meteorological factors that are physically relevant, such as wind speed, humidity, surface pressure, or convective potential, have not yet been incorporated. The exclusion of these features may limit the model's ability to capture complex atmospheric processes, particularly in scenarios driven by multi-variable interactions. Integrating additional variables, such as wind field structures or vertical profiles, into both the graph construction and node representation may enhance the model's responsiveness to high-impact weather systems.

Furthermore, although the model is generalizable to arbitrary geographic locations, our experiments have so far focused mainly on precipitation and temperature. Key atmospheric variables such as wind vectors or specific humidity have not yet been tested, and the current results do not fully assess the framework's applicability in broader multi-modal forecasting scenarios. Different atmospheric variables exhibit distinct spatial dependencies and temporal dynamics, and whether the current modeling paradigm generalizes across such diversity remains an open question.

The model also assumes the availability of some historical station-level observations. Although we avoid retraining large-scale backbones, the quality and availability of local data can still affect the performance of post-hoc corrections. In regions with highly sparse or irregular observations, such as new or mobile stations, the reliability of predictions may degrade. Enhancing model robustness through techniques such as transfer learning or structural priors could be a promising future direction.

Lastly, while the diffusion-based ensemble generation and selection mechanisms provide a degree of uncertainty modeling, the current framework does not explicitly incorporate physical consistency constraints. Soft constraints based on hydrological or thermodynamic conservation laws (e.g., moisture or energy balance) could improve both the interpretability and reliability of outputs, especially under extreme or anomalous conditions.

## C  Empirical Analysis of ERA5 Precipitation Interpolation Errors

To quantitatively motivate the need for high-resolution station-level correction, we conduct a systematic assessment of precipitation biases present in the ERA5 reanalysis dataset after standard bilinear interpolation onto surface meteorological stations.

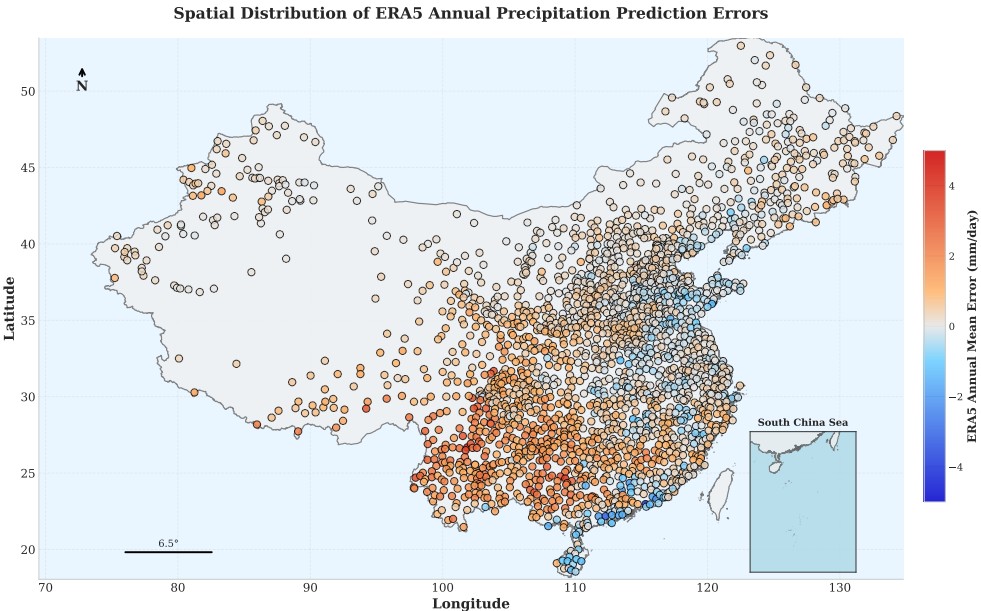

Figure 6: **Spatial Distribution of ERA5 Annual Precipitation Prediction Errors.** Each dot represents a station. Color denotes the signed mean error (in mm/day) of bilinearly interpolated ERA5 daily precipitation compared with ground observations, averaged over the entire year.

As shown in Figure 6, large spatial biases emerge across the region, with systematic overestimations in southwestern and southern China, and underestimations in southeastern coastal regions. These regional inconsistencies reflect the inability of grid-based interpolation to adapt to complex terrains and localized rainfall dynamics.

Figure 7 further illustrates the severity of these discrepancies in relative terms. In regions such as western Sichuan and Yunnan, the relative errors exceed 75%, suggesting that the interpolated ERA5 data may drastically misrepresent local annual precipitation totals. This disproportionate bias is particularly problematic in hydrologically sensitive regions or areas with sparse gauge density.

Figure 8 illustrates the temporal variation of interpolation errors in ERA5 precipitation data across the year. The top panel shows that both RMSE and mean error rise significantly during the summer months, indicating reduced accuracy during the rainy season. The middle panel reveals that the maximum and absolute maximum daily errors also spike during this period, with values occasionally exceeding 200 mm/day. The bottom panel shows that although many extreme rainfall events are

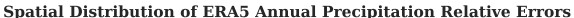

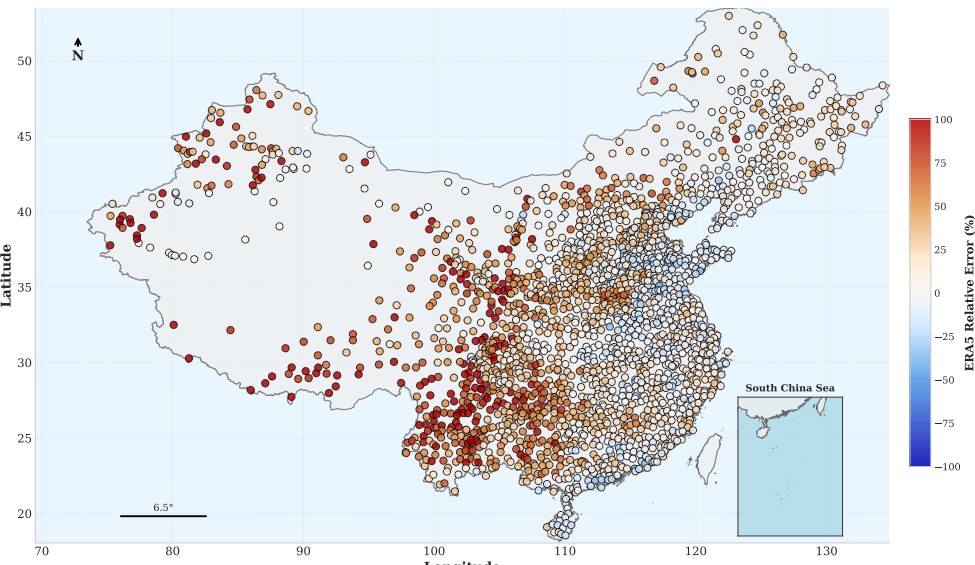

Figure 7: **Spatial Distribution of ERA5 Annual Precipitation Relative Errors.** The relative error is computed as the ratio between absolute bias and station-observed annual precipitation. High values indicate severe misrepresentation of total rainfall.

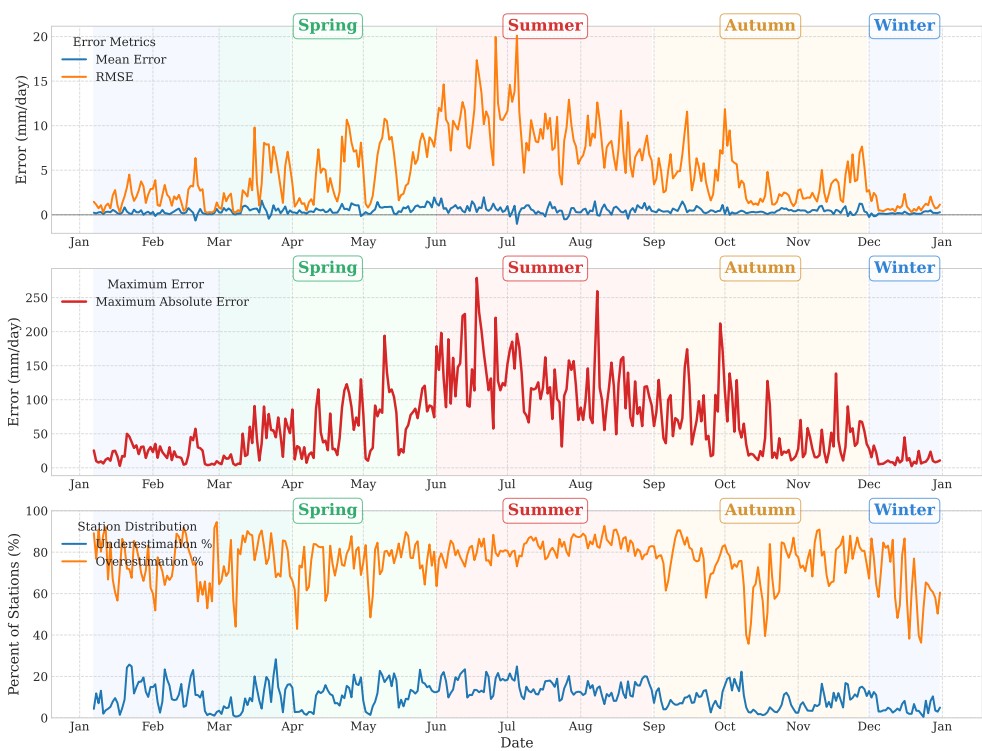

Figure 8: **Temporal Evolution of ERA5 Precipitation Interpolation Errors.** Top: daily root mean square error (RMSE) and mean error; Middle: maximum and maximum absolute errors; Bottom: percentage of stations with overestimation or underestimation. Shaded regions mark seasonal transitions.

underestimated in magnitude, a large proportion of stations still experience consistent overestimation throughout the year.

## D  Origins of Station Heterogeneity and Oversmoothing Biases

Despite the increasing availability of gridded reanalysis products such as ERA5, their direct interpolation to station locations often introduces non-trivial biases, particularly in regions with complex terrain or sparse observational coverage. These systematic errors give rise to two fundamental challenges in station-level precipitation modeling: *station heterogeneity* and *oversmoothing*.

**Station Heterogeneity.** As illustrated in Figures 6 and 7, bilinearly interpolated ERA5 precipitation fields exhibit highly uneven error distributions across space. Some stations experience persistent overestimation, while others are subject to substantial underestimation, particularly during periods of intense rainfall. This spatially inconsistent performance indicates that interpolation biases are inherently location-dependent and cannot be effectively corrected using globally shared model parameters.Importantly, this heterogeneity extends beyond spatial variability. As shown in Figure 8, interpolation accuracy also varies across seasons: errors are generally lower during dry periods, but escalate significantly during the rainy season. This seasonal inconsistency underscores the *temporal heterogeneity* of the interpolation task, where the statistical properties of precipitation—and the associated error patterns—change dynamically over time. Therefore, models that rely on static assumptions or uniform temporal behavior are likely to underperform in real-world applications. Capturing both spatial and temporal heterogeneity is essential for robust and generalizable interpolation.

**Oversmoothing of Extremes.** The temporal error profile shown in Figure 8 reveals that, although the overall interpolation error increases during the rainy season, the more critical failure lies in the underestimation of extreme rainfall events. In particular, the maximum error metrics in summer often exceed 200 mm/day, while the corresponding observed values are significantly higher. This pattern indicates that coarse-resolution gridded models tend to suppress sharp gradients and attenuate local extremes through spatial averaging. Such smoothing leads to the loss of physically meaningful high-magnitude signals—especially relevant for hydrological risk and extreme event forecasting. Consequently, downstream learning models trained on these smoothed inputs may underrepresent rare yet impactful weather patterns, limiting their utility in high-stakes applications.

Together, these findings highlight the empirical origins of the two central challenges tackled in this work. They motivate the design of interpolation frameworks that are not only spatially adaptive but also capable of preserving localized extremes. Subsequent sections build upon these insights to introduce model components that explicitly address heterogeneity and oversmoothing in a unified manner.

## E  Theoretical Analysis of OU-Based Temporal Encoding

To support the design of the GeoLiquidOU module, we provide a theoretical derivation of its Ornstein–Uhlenbeck (OU)-inspired update mechanism. This section analyzes how the model integrates historical input over time via exponential smoothing, and how its behavior corresponds to a first-order low-pass filter in the frequency domain.

### E.1  Continuous-Time OU Dynamics

We begin with the continuous-time OU process, excluding noise terms for clarity:

$$\frac{dh(t)}{dt} = \kappa \left[\theta(x(t)) - h(t)\right], \tag{4}$$

where $h(t)$ is the hidden state, $\theta(x(t))$ denotes an input-driven target signal, and $\kappa > 0$ is the decay rate.

Using an integrating factor $e^{\kappa t}$ and solving the ODE, we obtain:

$$h(t) = h(0)e^{-\kappa t} + \int_0^t \kappa \, e^{-\kappa(t-s)} \, \theta(x(s)) \, ds, \tag{5}$$

which reveals that the current state is a decayed memory of initial state plus an exponentially weighted integration of past inputs. The kernel $K(t) = \kappa e^{-\kappa t}$ acts as a memory filter with decay rate $\kappa$.

## E.2 Discrete-Time Approximation via Euler Method

To implement the update in discrete time with step size $\Delta t$, we apply Euler's method:

$$h[n+1] = h[n] + \Delta t \cdot \kappa \left[\theta[n] - h[n]\right]$$
$$= (1 - \alpha)h[n] + \alpha\theta[n], \tag{6}$$

where $\alpha = \kappa\Delta t \in (0,1)$. This update rule corresponds exactly to an Exponential Weighted Moving Average (EWMA), widely used in both time series filtering and biologically inspired computing.

## E.3 Explicit Recursion and Temporal Weighting

Unrolling the recursion across $N$ steps yields:

$$h[N] = (1-\alpha)^N h[0] + \sum_{i=0}^{N-1} \alpha(1-\alpha)^{N-1-i}\theta[i], \tag{7}$$

where the contribution of past inputs decays exponentially. The unnormalized weight for timestep $i$ is:

$$w_i = \alpha(1-\alpha)^{N-1-i}, \tag{8}$$

and the normalized weight becomes:

$$\bar{w}_i = \frac{w_i}{\sum_{j=0}^{N-1} w_j} = \frac{\alpha(1-\alpha)^{N-1-i}}{1 - (1-\alpha)^N}. \tag{9}$$

This shows that the discrete update implicitly applies a form of exponential memory over past $\theta$ values, where smaller $\alpha$ values yield longer memory horizons.

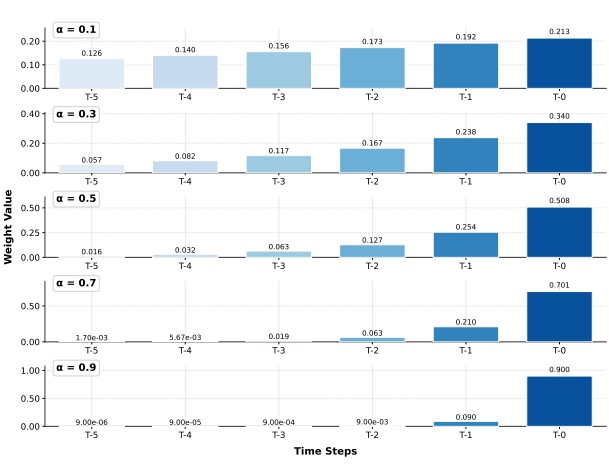

Figure 9: Visualization of OU attention weights.

## E.4 Frequency-Domain Interpretation: A Low-Pass Filter

We now analyze the above recursion as a linear time-invariant (LTI) system with input $\theta[n]$ and output $h[n]$. Applying the $z$-transform yields the transfer function:

$$H(z) = \frac{\alpha z^{-1}}{1 - (1-\alpha)z^{-1}}. \tag{10}$$

Evaluating on the unit circle $z = e^{i\omega}$ gives the magnitude response:

$$\left|H(e^{i\omega})\right| = \frac{\alpha}{\sqrt{1 - 2(1-\alpha)\cos\omega + (1-\alpha)^2}}. \tag{11}$$

This confirms that the system behaves as a classic first-order low-pass filter. Its key frequency characteristics are:

- **Zero frequency ($\omega = 0$)**:

$$|H(e^{i0})| = \frac{\alpha}{1 - (1-\alpha)} = 1.$$

  This indicates full preservation of low-frequency (DC) components, allowing long-term trends to pass through unchanged.

- **Highest frequency ($\omega = \pi$)**:

$$|H(e^{i\pi})| = \frac{\alpha}{\sqrt{(2-\alpha)^2}} = \frac{\alpha}{2-\alpha} \ll 1.$$

  This shows that high-frequency signals (rapid temporal variations) are strongly attenuated, effectively denoising short-term fluctuations.

- **General behavior**: The magnitude response $|H(e^{i\omega})|$ decreases monotonically with increasing $\omega$. The parameter $\alpha$ controls the cutoff: smaller values of $\alpha$ lead to longer memory and stronger smoothing.

### E.5 Interpretation for GeoLiquidOU

This analysis shows that the GeoLiquidOU module dynamically adjusts the temporal sensitivity of each station by learning a site-specific decay factor $\alpha = \tau(p) \cdot \kappa$, where $\tau(p)$ is a location-aware time constant. This enables the model to function as a learnable, spatially-conditioned low-pass filter, selectively integrating temporal context based on local variability.

## F  Latent-Space Noise Injection and UNet Denoising with Residual Perturbation

Let $z_0 = E(x_0) \in \mathbb{R}^d$ be the latent-space representation of the original interpolated value $x_0$. Under the DDPM framework [18, 36], the forward diffusion (noise injection) at step $t$ is defined by

$$z_t = \sqrt{\alpha_t}\, z_0 + \sqrt{1 - \alpha_t}\, \epsilon, \qquad \epsilon \sim \mathcal{N}(0, I_d),$$

so that

$$z_t \sim \mathcal{N}\big(\sqrt{\alpha_t}\, z_0,\ (1 - \alpha_t)I_d\big).$$

Here, $\alpha_t \in (0, 1)$ denotes the precomputed noise-schedule coefficient.

In the reverse (denoising) step, a UNet model $\epsilon_\theta(z_t, t)$ is trained to predict the injected noise. Denote the prediction error by

$$\delta = \epsilon_\theta(z_t, t) - \epsilon,$$

which can be modeled as a zero-mean Gaussian random vector with covariance $\sigma_\delta^2 I_d$. The corresponding one-step reconstruction in latent space is then

$$\tilde{z}_0 = \frac{1}{\sqrt{\alpha_t}}\Big(z_t - \sqrt{1 - \alpha_t}\, \epsilon_\theta(z_t, t)\Big) = z_0 - \underbrace{\frac{\sqrt{1 - \alpha_t}}{\sqrt{\alpha_t}}\, \delta}_{\Delta z}.$$

Since $\delta \sim \mathcal{N}(0, \sigma_\delta^2 I_d)$, it follows that

$$\tilde{z}_0 \sim \mathcal{N}\Big(z_0,\ \tfrac{1 - \alpha_t}{\alpha_t}\, \sigma_\delta^2\, I_d\Big).$$

Because the residual covariance $\frac{1 - \alpha_t}{\alpha_t} \sigma_\delta^2 I_d$ is positive definite, the reconstructed latent vector $\tilde{z}_0$ admits nonzero variance around $z_0$. In other words, despite denoising, the model retains a controlled stochastic perturbation in latent space, which—when subsequently decoded—enables recovery or amplification of extreme values beyond the initial interpolated range.

# G Cross-Variable Forecasting: Temperature as Input

To verify the adaptability of our model architecture to alternative meteorological variables, we conduct a parallel experiment using 2-meter temperature (T2M) fields from **Pangu-Weather**. While the entire model is retrained from scratch on the T2M dataset, the network structure remains unchanged.

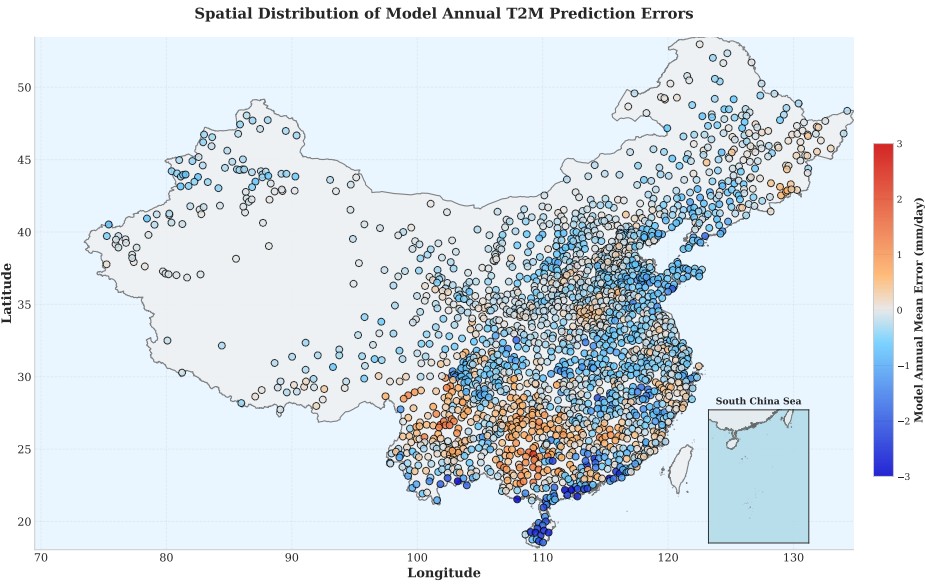

Figure 10: **Spatial Distribution of Absolute Temperature Errors.** The annual mean absolute error (MAE) of 2-meter temperature predictions across China using our method **DiffLiG**.

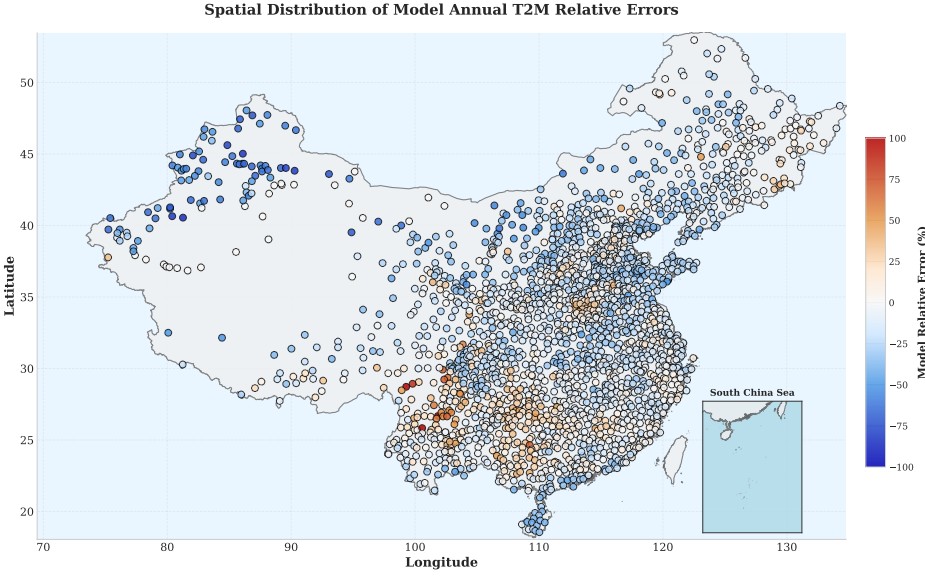

Figure 11: **Spatial Distribution of Relative Temperature Errors.** The relative error percentage (%) of 2-meter temperature predictions, highlighting spatial performance differences across varied terrain and climate zones.

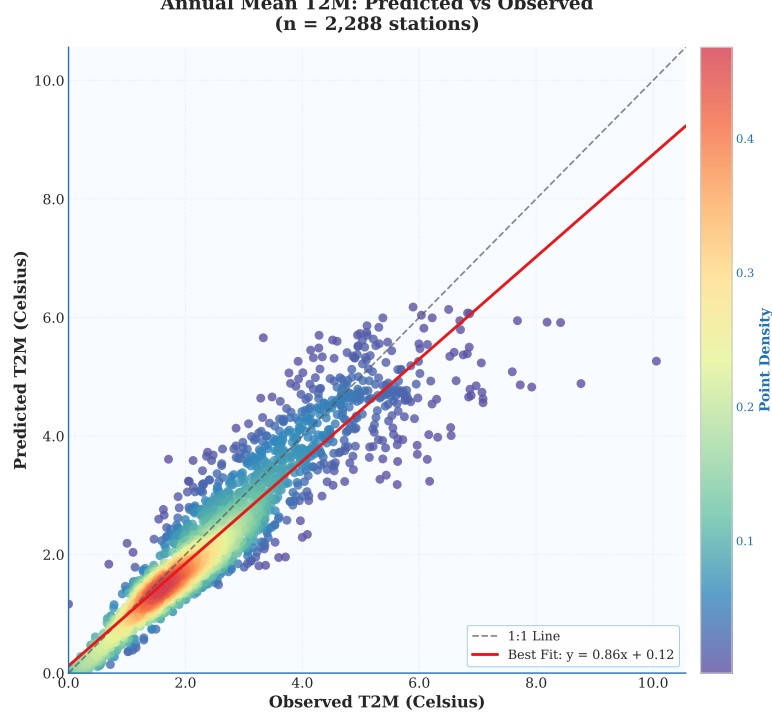

Figure 12: **Scatter Plot of Predicted vs Observed Annual Mean T2M.** This plot shows station-level comparisons between predicted and observed annual average temperatures. A best-fit line ($y = 0.86x + 0.12$) is provided alongside the identity line.

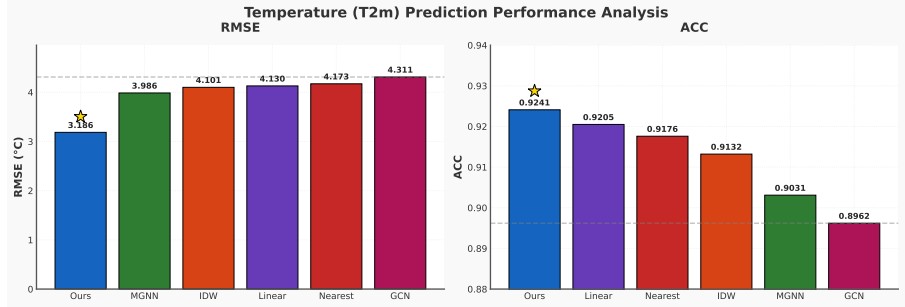

Figure 13: **Performance Comparison on 2-meter Temperature Prediction.** Our method **DiffLiG**, retrained using Pangu-Weather T2M inputs, achieves the best RMSE and ACC scores among baseline methods including MGNN, GCN, and classical interpolation techniques.

# H  Sensitivity to Forecast Horizon and Graph Connectivity

To evaluate the robustness and stability of our model across different forecasting and structural configurations, we conduct a series of sensitivity experiments. Specifically, we examine the impact of four critical parameters:

- **Forecast lead time**,
- **Retrospective window size**,
- **Maximum number of edges from ERA5 grid points to stations**
- **Maximum number of inter-station connections**.

Unless otherwise specified, the model is trained with the following default configuration: forecast lead time of **1 day**, retrospective window size of **5 days**, a maximum of **4 connections** from each ERA5 grid point to nearby stations, and up to **3 connections** between observation stations. The goal of this analysis is to assess the sensitivity of model performance with respect to these parameters, thereby providing insight into the design choices behind spatial and temporal coupling strategies.

## H.1  Sensitivity to Forecast Lead Time

Figure 14: **Impact of Lead Time on Forecast Accuracy.**

To assess how prediction horizon affects model performance, we conduct a sensitivity analysis by varying the *lead time* from 1 to 7 days. As shown in Figure 14, the model retains relatively high accuracy and low RMSE at short-term forecasts. However, performance deteriorates as the prediction window extends, particularly in terms of event detection (TS@2mm). This suggests that while the

model can extrapolate coarse patterns over longer horizons, precise localization of extreme events becomes more challenging.

## H.2 Impact of Backtracking Window Size

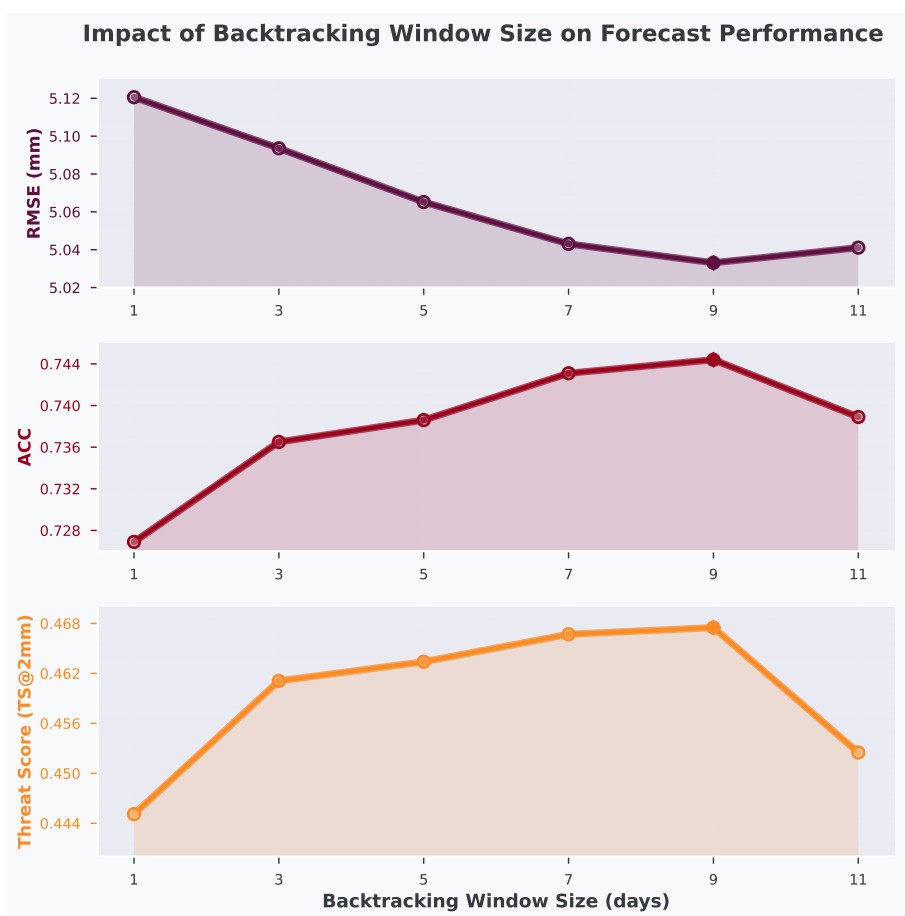

Figure 15: **Forecast performance under different temporal backtracking window sizes.**

Figure 15 illustrates how the backtracking window size—i.e., the number of past days used for station-level temporal encoding—affects forecast performance. As the window length increases, performance steadily improves across all metrics, suggesting that a longer temporal context helps capture the heterogeneous evolution of local precipitation patterns. To balance accuracy and computational efficiency, we choose a 5-day window as the default configuration.

## H.3 Impact of ERA5-to-Station Edge Connectivity

Figure 16 illustrates the impact of varying the maximum number of ERA5-to-station edges on prediction performance. As the number of edges increases, we observe consistent improvements across RMSE. However, the performance gain plateaus beyond 4 edges, indicating diminishing returns with denser spatial connections. Based on this trend, we select 4 as the default edge limit in our model configuration to balance accuracy and computational efficiency.

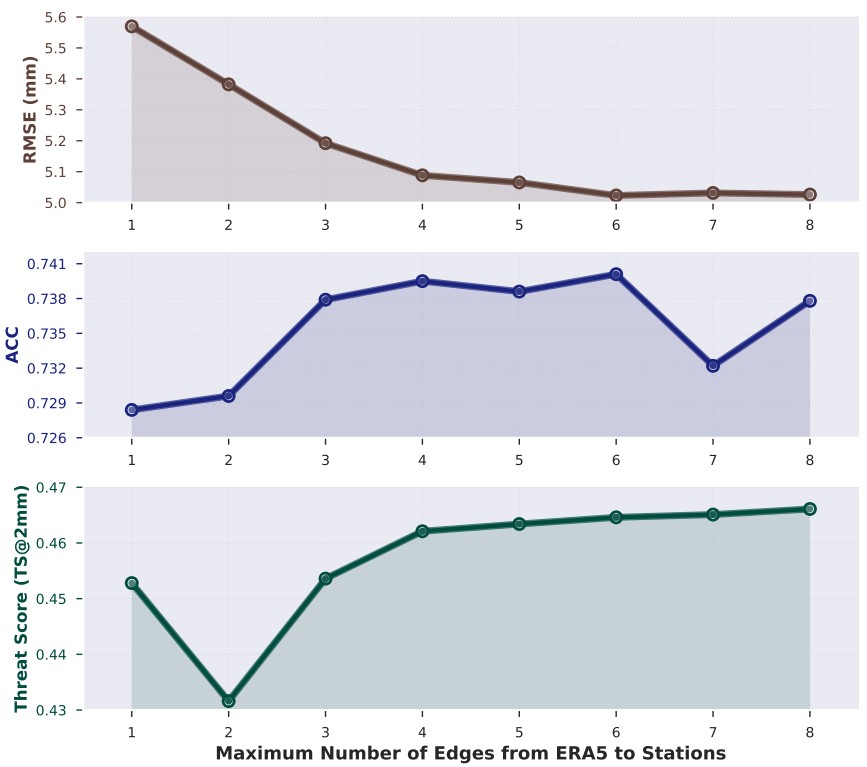

Figure 16: **Forecast performance across different ERA5-to-station edge limits.**

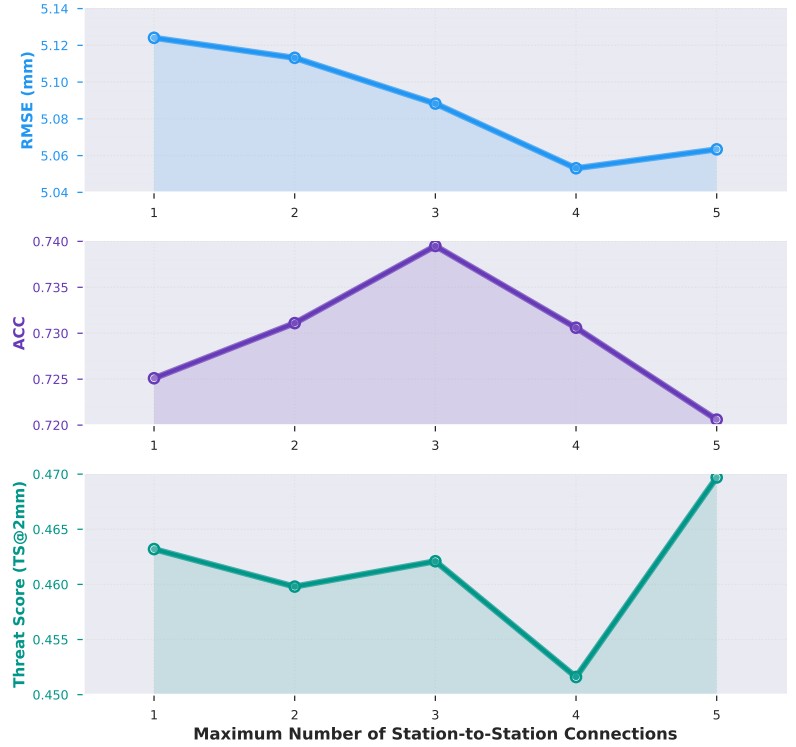

Figure 17: **Forecast performance under different inter-station connectivity levels.**

### H.4 Impact of Station-to-Station Connectivity

Figure 17 shows the effect of varying the number of station-to-station connections on forecast performance. Unlike the ERA5-to-station connectivity, increasing inter-station edge counts leads to only minor fluctuations in RMSE, ACC, and TS@2mm. The performance remains relatively stable across different settings, indicating that the model is less sensitive to this parameter. We therefore fix the maximum number of inter-station edges at 3 for subsequent experiments.

# I  Implementation and Training Details

## I.1  Training Objective and Loss Function

The proposed graph-based forecasting framework is trained in a supervised regression setting. The objective is to minimize the discrepancy between the predicted values and ground-truth observations at meteorological stations, thereby enabling accurate station-level interpolation.

Let $\hat{y}_i$ denote the predicted value and $y_i$ the corresponding ground-truth observation for the $i$-th sample across time and space. The loss function is defined as the mean squared error (MSE) over all station samples:

$$\mathcal{L}_{\text{MSE}} = \frac{1}{N} \sum_{i=1}^{N} \|\hat{y}_i - y_i\|^2 \tag{12}$$

where $N$ is the total number of training samples. This loss directly supervises the end-to-end prediction pipeline, encouraging the model to align with observed precipitation patterns across both spatial and temporal dimensions.

## I.2  Task Configuration

We formulate the task as a station-level spatial interpolation problem, where the model observes past $T_1 = 5$ days of gridded data to predict the station value at the next day ($T_2 = 1$). The temporal granularity is set to daily. For each target station, the maximum number of spatial edges is constrained to ensure computational efficiency and physical relevance:

- **Grid-to-station connections:** at most 4 ERA5 grid nodes;
- **Station-to-station connections:** at most 3 surrounding stations.

We use a 7-year dataset ranging from 2016 to 2022. Specifically, we assign the years 2016–2020 for training, 2021 for validation, and 2022 for final testing. All models are implemented using the PyTorch deep learning framework.

## I.3  Training Configuration

The entire training process is conducted on a single NVIDIA RTX 4090D GPU with 24GB of memory. Each training epoch requires approximately 3 minutes. The total training spans 150 epochs and completes in about 7.5 hours. This setup demonstrates strong computational efficiency and scalability.

## I.4  Staged Training Strategy

We progressively activate key modules to stabilize training dynamics and ease inter-module adaptation. The training process follows the table 7:

# J  Haversine Distance Computation

To measure the great-circle distance between two stations on the Earth's surface, we employ the Haversine formula, which accounts for the spherical geometry of the planet.

Let the two geographic locations be denoted as $\mathbf{p}_i = (\lambda_i, \phi_i)$ and $\mathbf{p}_j = (\lambda_j, \phi_j)$, where $\lambda$ and $\phi$ represent the longitude and latitude (in degrees), respectively. These coordinates are first converted into radians:

Table 6: Training Hyperparameters

| Parameter | Value |
|---|---|
| Framework | PyTorch |
| Optimizer | Adam [20] |
| Initial Learning Rate | 1e-4 |
| Weight Decay | 1e-4 |
| Batch Size | 16 |
| Epochs | 150 |
| Time Window | $T_1 = 5$ (past), $T_2 = 1$ (future) |
| Grid Edges (max) | 4 |
| Station Edges (max) | 3 |
| Loss Function | MSE Loss |
| Dropout | 0.1 |
| Activation | LeakyReLU / DynamicTanh [43] |
| Training Years | 2016–2020 |
| Validation Year | 2021 |
| Test Year | 2022 |

Table 7: Staged Training Strategy

| Stage | Epoch Range | Activated Modules |
|---|---|---|
| Warm-up | 0–20 | Basic GNN backbone with fixed topology |
| GNN Training | 20–40 | Activate Dynamic Edge Modulator |
| Forecast Diversity | 40–60 | Enable Probabilistic Forecast Diffuser |
| Selector Adaptation | 60–80 | Introduce Multi-Criteria Score Selector |
| Station-wise Bias | 80–100 | Learn station-specific bias correction |
| Joint Training | 100–150 | Fine-tune all modules jointly for synergy |

$$\lambda_i^{\mathrm{rad}} = \lambda_i \cdot \frac{\pi}{180}, \quad \phi_i^{\mathrm{rad}} = \phi_i \cdot \frac{\pi}{180} \tag{13}$$

We then compute the difference in latitude and longitude:

$$\Delta\phi = \phi_j^{\mathrm{rad}} - \phi_i^{\mathrm{rad}}, \quad \Delta\lambda = \lambda_j^{\mathrm{rad}} - \lambda_i^{\mathrm{rad}} \tag{14}$$

The Haversine formula estimates the spherical distance $d_{ij}$ between the two points as follows:

$$a = \sin^2\left(\frac{\Delta\phi}{2}\right) + \cos(\phi_i^{\mathrm{rad}}) \cdot \cos(\phi_j^{\mathrm{rad}}) \cdot \sin^2\left(\frac{\Delta\lambda}{2}\right) \tag{15}$$

$$c = 2 \cdot \arctan 2\left(\sqrt{a}, \sqrt{1-a}\right) \tag{16}$$

$$d_{ij} = R \cdot c \tag{17}$$

Here, $R = 6371$ km denotes the average radius of the Earth.

# K Baseline Method Descriptions

To evaluate the effectiveness and generalizability of our proposed framework, we compare against a broad set of baseline models, categorized into four methodological groups. These baselines cover a range of paradigms from classical statistical interpolation to advanced neural forecasting models.

**(1) Traditional Interpolation Methods.** We include four classic spatial interpolation techniques widely used in geostatistics and meteorology:

- **Nearest Neighbor** assigns each station the value of the closest grid point without smoothing, often leading to abrupt spatial transitions.

- **Linear Interpolation** assumes uniform variation between nearby points, but does not capture spatial correlation or terrain effects.

- **Inverse Distance Weighting (IDW)** assigns weights inversely proportional to distance, introducing smooth decay but lacking adaptivity to local density or directional features.

- **Ordinary Kriging (OK)** estimates station values using spatial covariance models (variograms), providing statistically grounded predictions but requiring stationarity assumptions and manual variogram fitting.

While these methods are computationally efficient and interpretable, they rely on static spatial assumptions, lack learnable parameters, and do not account for temporal context or spatial heterogeneity, resulting in degraded performance over complex terrains or data-sparse regions.

**(2) Graph Neural Network (GNN) Models.**   GNNs provide a flexible framework for modeling spatial relationships among stations by learning representations through message passing:

- **GCN** [19] applies localized spectral convolutions on fixed station graphs, but tends to oversmooth signals across the network.

- **GAT** [38] introduces attention mechanisms to weigh neighbor importance, improving expressiveness but still relying on static edges.

- **GraphSAGE** [14] supports inductive learning via sampled aggregation, yet assumes homogeneous information flow across all regions.

- **KCN** [3] combines kernel-based weighting with graph convolutions to capture geostatistical structure, but uses fixed kernels and non-adaptive connectivity.

Despite their spatial modeling strengths, these GNNs often employ fixed topologies and uniform propagation rules, which limit their ability to adjust to station-specific characteristics and introduce oversmoothing in high-resolution prediction tasks.

**(3) Physics-Inspired and Multimodal Graphs.**   We also include two architectures that incorporate physical priors or multimodal inputs:

- **MeshGraphNet** [31] learns graph-based simulators aligned with physical PDEs, capturing spatial continuity but requiring large data and stable mesh structures.

- **MGNN** [39] constructs heterogeneous graphs combining reanalysis grids and observational stations, enabling multimodal fusion but still relying on predefined graph rules and lacking dynamic edge adaptation.

These models offer better spatial fidelity than standard GNNs but often inherit rigid structures or handcrafted designs, limiting flexibility in highly heterogeneous or evolving forecast settings.

**(4) Generative and Transformer-Based Spatial Models.**   To assess uncertainty modeling and spatial diversity, we include:

- **Diffusion Models** [18] generate samples via iterative noise reversal, allowing diverse outputs but often requiring long sampling chains and lacking explicit graph structure.

- **Vision Transformer (ViT)** [11] applies global self-attention over spatial patches, capturing long-range dependencies but prone to global averaging, which may obscure local extremes or abrupt transitions.

These models demonstrate strong generative capacity, yet often lack station-aware structure and may smooth over localized variations, making them less suitable for fine-grained, heterogeneous interpolation tasks.

Together, these baselines form a representative and challenging benchmark suite, allowing us to assess not only interpolation accuracy but also spatial adaptivity, generalization capacity, and robustness to heterogeneity.

# L    Evaluation Metrics

We adopt three widely used metrics to evaluate interpolation and prediction accuracy for station-level precipitation: Anomaly Correlation Coefficient (ACC), Root Mean Square Error (RMSE), and Threat Score (TS) under a 2 mm threshold. The formal definitions are provided below.

Given the true values $y_i$ and predicted values $\hat{y}_i$ at each location $i$, we define their respective climatological means $\bar{y}_i$ and $\bar{\hat{y}}_i$ as the multi-year averages over a reference period for the observed and predicted data, respectively. The anomaly correlation coefficient (ACC) is then computed as:

$$\text{ACC} = \frac{\sum_i (y_i - \bar{y}_i)(\hat{y}_i - \bar{\hat{y}}_i)}{\sqrt{\sum_i (y_i - \bar{y}_i)^2} \cdot \sqrt{\sum_i (\hat{y}_i - \bar{\hat{y}}_i)^2}} \tag{18}$$

ACC measures the pattern similarity between predicted and observed anomalies, accounting for inter-location variance.

**Root Mean Square Error (RMSE).** RMSE evaluates the overall deviation between predictions and ground truth, and is defined as:

$$\text{RMSE} = \sqrt{\frac{1}{N} \sum_{i=1}^{N} (y_i - \hat{y}_i)^2} \tag{19}$$

where $N$ is the total number of evaluated samples. RMSE is sensitive to large errors and commonly used in regression tasks.

**Threat Score at 2mm (TS@2mm).** To assess the model's ability to detect light to moderate precipitation events, we calculate the Threat Score (TS) under a fixed threshold of 2 mm/day:

$$\text{TS@2mm} = \frac{\text{TP}}{\text{TP} + \text{FN} + \text{FP}} \tag{20}$$

where TP (true positives), FN (false negatives), and FP (false positives) are determined based on thresholded binary classification:

$$\text{TP} = \sum_i \mathbf{1}[y_i \geq 2 \wedge \hat{y}_i \geq 2], \quad \text{FN} = \sum_i \mathbf{1}[y_i \geq 2 \wedge \hat{y}_i < 2], \quad \text{FP} = \sum_i \mathbf{1}[y_i < 2 \wedge \hat{y}_i \geq 2]$$

TS reflects the model's event-level accuracy in a binary detection task and is widely used in precipitation forecast verification.

# M    License and Attribution Details

This study uses several existing assets whose licenses and usage terms are properly acknowledged and respected:

- **Model License.** The proposed model will be released under the CC-BY 4.0 license upon publication.

- **Baselines.** All baseline models used in our experiments are implemented based on open-source releases from their original papers. We cite the corresponding works in our main text, and all implementations respect the original licenses provided by their authors.

- **ERA5 Data.** We use the hourly precipitation product from the ERA5 reanalysis dataset, provided by the European Centre for Medium-Range Weather Forecasts (ECMWF). This dataset is publicly accessible via the [Copernicus Climate Data Store](https://cds.climate.copernicus.eu/) and is licensed under CC-BY 4.0. We use the version released in 2022.

- **CMA Observation Data.** The ground truth observation data from the China Meteorological Administration (CMA) is not publicly available. Access to this dataset was granted to us through proper institutional authorization.

No scraped, re-packaged, or third-party datasets outside of these sources are used in this study.

# N Visual comparison of spatial interpolation results

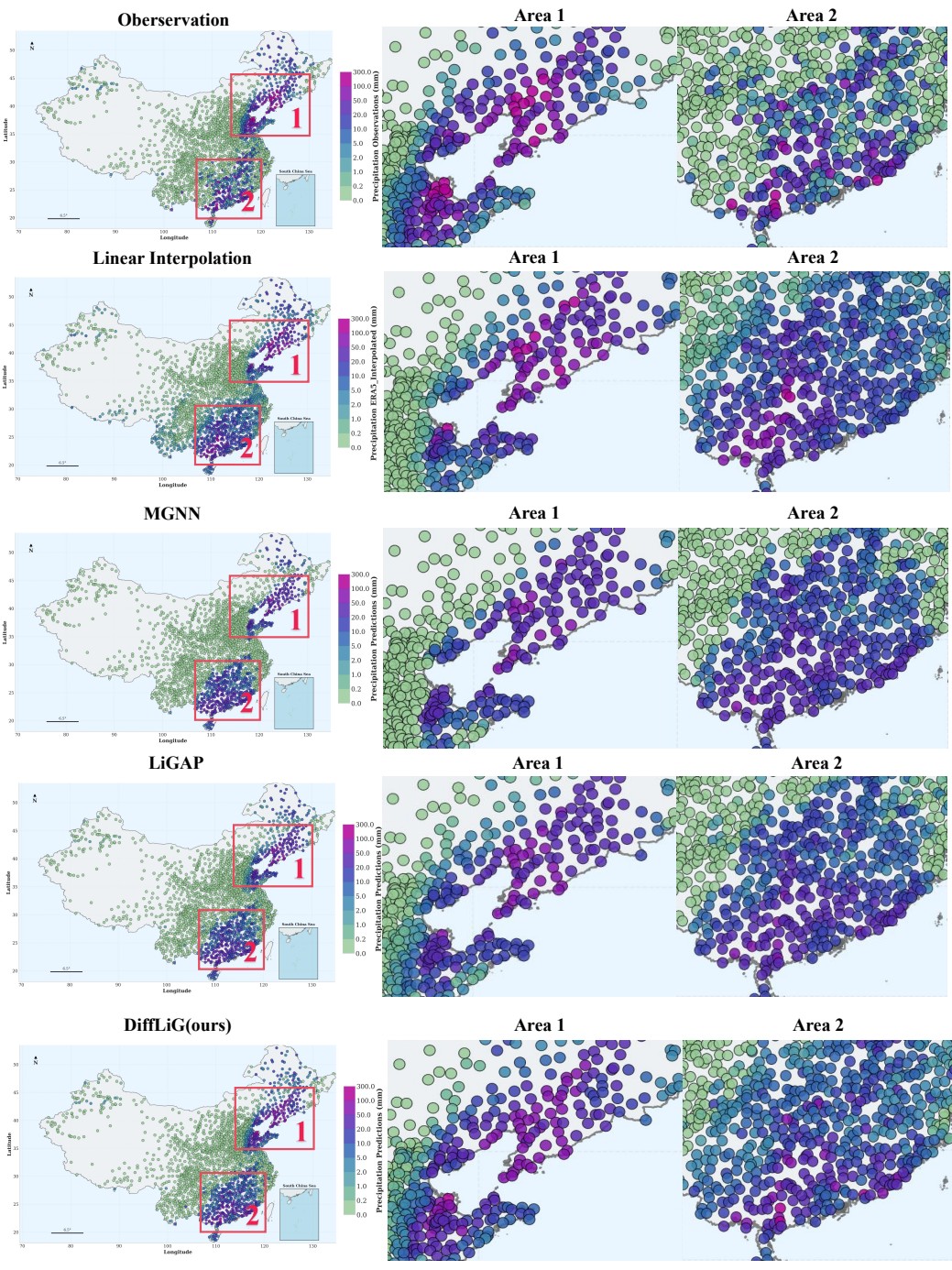

Figure 18: **Visual comparison of spatial interpolation results.** Predicted daily accumulated precipitation for July 5, 2022.

