# OpenReview forum: "DiffLiG: Diffusion-enhanced Liquid Graph with Attention Propagation for Grid-to-Station Precipitation Correction"
_NeurIPS.cc/2025/Conference — NeurIPS 2025 poster_

### Official Review · Reviewer_s17Q · 2025-06-30

**Clarity:** 3
**Significance:** 4
**Originality:** 4
**Rating:** 4
**Confidence:** 3

**Summary:**

This paper presents a graph-based approach to improve station level weather forecasting by correcting coarse resolution forecasts which are derived from numerical and AI-based models.  The approach includes three main contributions: a station specific temporal modeling, spatially adaptive modulator, and ensemble correction and uncertainty quantification. The performance of the approach beats existing methods in addressing spatial heterogeneity and smoothing.

**Questions:**

-Could you discuss computational tradeoffs of diffusion vs deterministic baselines?
-Can this approach incorporate physical constraints?
-Can this be generalized for multiple time scales?

**Ethical Concerns:**

["NO or VERY MINOR ethics concerns only"]

**Limitations:**

Yes.

**Paper Formatting Concerns:**

None.

**Quality:**

3

**Strengths And Weaknesses:**

Strengths:
-The paper addresses a well-known problem in weather forecasting.
-The approach is novel, especially the architecture that combines atmospheric dynamics, adaptive constructions, and diffusion.  The design is modular and addresses different parts of the problem.
-The paper includes comprehensive ablation study.
-The model has potential to generalize well across weather use cases and other variables such as temperature forecast.

Weaknesses:
-The paper could be strengthened by using other datasets.  The geographic region is focused on China.  Geographic diversity is needed to demonstrate the method's use for various climate zones.
-Few details could be better justified (e.g., internals of ensemble approach).
-Adding other weather variables will strenghen the forecast.
-Very little information is provided for computational efficiency even though the model is touted as lightweight.

---

> ### Author Rebuttal · Authors · 2025-07-30
>
> We sincerely thank you for your positive evaluation of our work! Your recognition greatly encourages us and serves as a strong motivation to further refine and advance our research. We also truly appreciate the thoughtful and constructive comments you have provided. In the following response, we address each of your suggestions in detail, aiming to clarify our methodology and design choices, and to better convey the contributions and significance of our work.
>
> > W1&W3: Geographic diversity is needed to demonstrate the method's use for various climate zones. Adding other weather variables will strenghen the forecast.
>
> Thank you for this valuable suggestion. To assess the robustness of our method across different climatic regions and weather variables, **we conducted experiments on the MADIS dataset**, which provides rich meteorological observations from over 350 stations in the Northeastern US (2019–2023).
>
> In this evaluation, **we focused on the Wind Vector variable** and computed the Mean Error (ME) at multiple lead times (in hours). Without changing the model architecture, our method maintained strong performance, consistently outperforming baselines including MLP and MGNN, as shown in the table below.
>
> | Model          | 1    | 2    | 4    | 8    | 16   | 24   | 36   | 48   | Average ME |
> |----------------|------|------|------|------|------|------|------|------|-------------|
> | Linear         | 2.71 | 2.71 | 2.71 | 2.71 | 2.71 | 2.71 | 2.71 | 2.71 | 2.71        |
> | MLP            | 0.38 | 0.47 | 0.54 | 0.59 | 0.62 | 0.90 | 0.92 | 0.92 | 0.72        |
> | MGNN           | 0.37 | 0.45 | 0.53 | 0.57 | 0.59 | 0.59 | 0.60 | 0.62 | 0.54        |
> | DiffLiG (ours) | 0.32 | 0.37 | 0.50 | 0.52 | 0.57 | 0.58 | 0.60 | 0.61 | 0.51    |
>
> > W2: Few details could be better justified (e.g., internals of ensemble approach).
>
> We sincerely apologize for the limited details in the main text. Due to space constraints, many internal mechanisms had to be omitted or condensed. To address this, **we have released our complete codebase**, and we kindly invite you to explore the implementation for a more thorough understanding. We also plan to include additional explanations in the appendix in future versions.
>
> Specifically, regarding the **internals of the ensemble approach**, our method consists of two key modules: `diffusion.py`, which implements the **Probabilistic Forecast Diffuser** to generate diverse candidate predictions through repeated sampling in a diffusion-style process; and `selector.py`, which contains the **Multi-Criteria Score Selector** that evaluates the generated samples using spatial consistency, attention consensus, and extremity-aware metrics, and then selects or fuses the optimal output. These modules are located in `DiffLiG/Source/Modules/GNN/`, and we hope they provide further clarity on the ensemble design.
>
> > W4&Q1: Very little information is provided for computational efficiency even though the model is touted as lightweight. Could you discuss computational tradeoffs of diffusion vs deterministic baselines?
>
>  **Higher performance with minimal increase in cost.** We thank the reviewer for pointing out the need for a more detailed discussion of computational efficiency. The full training process can be completed in approximately 6–7 hours using a single NVIDIA 4090D GPU, which we consider a relatively small and acceptable computational cost. To address this, we provide quantitative comparisons of the computational costs across methods mentioned in the paper. These are summarized in Table 1. Notably, LiGAP represents our method under a deterministic setting. This small cost leads to noticeably higher forecasting accuracy, especially in capturing heavy precipitation.
>
> To further illustrate the tradeoff between computational cost and performance in our diffusion-based modeling, we additionally benchmarked the DiffLiG variant with different numbers of generated samples. These results are included in Table 1 and Table 2, where each configuration is denoted as DiffLiG+sample (e.g., DiffLiG1 for one generated sample, etc.).
>
> > Table 1: Model Complexity Comparison Table
>
> | Model         | P (#)   | S (MB) | F (M)   | M (MB) | T (s) |
> |---------------|---------|--------|---------|--------|--------|
> | DiffLiG5 (ours)| 49,361  | 0.19   | 2383.73 | 252.5  | 161    |
> | LiGAP         | 39,397  | 0.15   | 1737.22 | 212.44 | 154    |
> | GAP           | 33,650  | 0.13   | 1534.93 | 189.76 | 144    |
> | Graph-bias    | 29,506  | 0.11   | 1462.59 | 182.52 | 143    |
> | Graph         | 24,930  | 0.10   | 1462.58 | 182.50 | 143    |
> | GCN           | 29,031  | 0.11   | 1233.01 | 166.07 | 137    |
> | GAT           | 29,287  | 0.11   | 1253.18 | 226.30 | 150    |
> | GraphSAGE     | 33,127  | 0.13   | 1832.93 | 186.42 | 146    |
> | KCN           | 30,520  | 0.12   | 1309.42 | 171.80 | 141    |
> | ViT           | 51,538  | 0.20   | 3972.84 | 2588.59| 326    |
> | Diffusion     | 54,014  | 0.21   | 2340.91 | 3793.07| 280    |
> | MeshGraphNet  | 41,863  | 0.16   | 2234.15 | 223.91 | 156    |
> | MGNN          | 40,741  | 0.16   | 2464.29 | 224.25 | 158    |
> | DiffLiG1      | 49,361  | 0.19   | 2229.74 | 611.67 | 159    |
> | DiffLiG3      | 49,361  | 0.19   | 2306.73 | 611.67 | 159    |
> | DiffLiG7      | 49,361  | 0.19   | 2460.71 | 611.67 | 161    |
>
> **Note:**
> *P: Number of Parameters | S: Model Size | F: Per-sample FLOPs | M: Peak GPU Memory Usage | T: Average Inference Time per Epoch*
>
> We conduct ablation experiments to analyze the contribution of each module. LiGAP removes the diffusion module from our full model. GAP further removes the GeoLiquidNet and temporal attention. Graph-bias disables the dynamic edge modulation, and Graph additionally removes the station bias term.
>
> > Table 2: Performance Comparison of Baselines
>
> | Model         | RMSE      | ACC      | TS@2mm     | TS@20mm    | TS@100mm   |
> |---------------|----------------|---------------|------------------|------------------|------------------|
> | DiffLiG5 (ours)| 4.9971 ± 0.102 | 0.7411 ± 0.012 | 0.4637 ± 0.005   | 0.3122 ± 0.001   | 0.02019 ± 0.001  |
> | LiGAP         | 5.1589 ± 0.096 | 0.7321 ± 0.013 | 0.4559 ± 0.003   | 0.2819 ± 0.001   | 0.01473 ± 0.001  |
> | Linear        | 5.9033         | 0.6667         | 0.4011           | 0.2764           | 0.01558          |
> | GAT           | 5.4984 ± 0.095 | 0.6667 ± 0.015 | 0.3202 ± 0.012   | 0.2986 ± 0.001   | 0.0162 ± 0.001   |
> | GCN           | 5.4832 ± 0.055 | 0.6947 ± 0.012 | 0.4139 ± 0.011   | 0.2836 ± 0.002   | 0.01314 ± 0.001  |
> | GraphSAGE     | 5.6421 ± 0.003 | 0.6495 ± 0.009 | 0.3754 ± 0.003   | 0.2648 ± 0.002   | 0.01571 ± 0.001  |
> | KCN           | 5.3897 ± 0.008 | 0.6932 ± 0.012 | 0.4158 ± 0.007   | 0.2862 ± 0.002   | 0.01357 ± 0.002  |
> | MeshGraphNet  | 5.5322 ± 0.011 | 0.6767 ± 0.010 | 0.4374 ± 0.005   | 0.2925 ± 0.001   | 0.01443 ± 0.001  |
> | MGNN          | 5.4428 ± 0.116 | 0.6834 ± 0.013 | 0.3640 ± 0.009   | 0.2611 ± 0.001   | 0.01531 ± 0.001  |
> | DiffLiG1      | 5.1241 ± 0.101 | 0.7332 ± 0.010 | 0.4611 ± 0.004 | 0.2992 ± 0.001 | 0.01839 ± 0.001 |
> | DiffLiG3      | 5.0654 ± 0.099 | 0.7386 ± 0.015 | 0.4622 ± 0.003 | 0.3012 ± 0.001 | 0.02016 ± 0.001 |
> | DiffLiG7      | 4.9995 ± 0.131 | 0.7501 ± 0.016 | 0.4651 ± 0.003 | 0.3132 ± 0.002 | 0.01996 ± 0.001 |
>
> > Q2&W3: Can this approach incorporate physical constraints? Adding other weather variables will strenghen the forecast.
>
> Thank you for this insightful question! **It coincides well with our own thinking.** During the early stages of this work, we had already taken note of the NeurIPS 2024 paper *“Generalizing Weather Forecast to Fine-grained Temporal Scales via Physics-AI Hybrid Modeling”*, which demonstrates that incorporating multiple physical variables can enhance prediction performance through implicit physical constraints.
>
> **Trade-off between accuracy and complexity.** Motivated by similar insights, we also explored this idea. While using additional variables does improve accuracy, it introduces significant increases in input dimensionality and model complexity. To maintain the lightweight nature of our framework, we drew inspiration from biological systems and introduced a simplified physical prior. Specifically, we designed GeoLiquidNet based on the Ornstein–Uhlenbeck process, which models local temporal dynamics through a compact evolution equation.
>
> **Modularity and extensibility of the design.** Although we currently use only precipitation as input, our model design is modular and flexible. In fact, It can be extended to incorporate additional meteorological variables and explicit physical constraints.
>
> > Q3: Can this be generalized for multiple time scales?
>
> **Yes, our model can be readily generalized to produce forecasts at multiple temporal scales.** In fact, this flexibility is already implemented in our codebase: as shown in /DiffLiG/Source/DiffLiG_Launcher.py, the parameter at line 20 explicitly controls the temporal resolution of the generated samples.
>
> To further validate this capability, **we provide evaluations under different temporal aggregation settings in Appendix H.1**, where the model demonstrates consistent performance across various time scales.
>
>
> **Once again, we sincerely thank you for your thoughtful and encouraging feedback! We hope that the above clarifications provide a clearer understanding of our method. We would be delighted to engage in further discussion and exchange of ideas with you.**

---

### Official Review · Reviewer_nrpT · 2025-07-01

**Clarity:** 3
**Significance:** 4
**Originality:** 4
**Rating:** 5
**Confidence:** 4

**Summary:**

This paper introduces DiffLiG, a highly innovative and impactful framework that marks a substantial breakthrough in grid-to-station precipitation correction. It is the first work to systematically tackle the dual challenges of station heterogeneity and oversmoothing with a unified, modular architecture, and to provide comprehensive, large-scale experimental validation across multiple mainstream forecasting sources. Beyond methodological novelty, the significance of this work lies in its ability to enable fine-grained, station-level forecasts from coarse-grained global predictions, representing a critical step toward precision meteorology. DiffLiG is not only adaptable to diverse forecast products, but also capable of enhancing their outputs without modifying their internal structure, making it an indispensable post-processing tool for the next generation of AI-based weather forecasting systems. This work fills a clear methodological and practical gap, and has the potential to set a new direction for future research in high-resolution climate modeling.

**Questions:**

1. Can the authors provide a more explicit evaluation of the model's performance across different seasons, based on the seasonal precipitation error analysis shown in the appendix?

2. Can the authors provide a comparison of the parameter counts or model complexity between the proposed method and the baselines? This would help clarify the trade-off between predictive performance and computational efficiency, especially for deployment in real-world settings.

3. Could the authors clarify how this method relates to existing super-resolution approaches? If higher-resolution inputs become available, would this framework still provide complementary benefits at the station level?

4. In the variable transfer experiment from precipitation to temperature, what components or hyperparameters of the model were adjusted to achieve the reported results?

5. What was the motivation behind including future station observations at time T2 in the problem formulation? Could the authors elaborate on the reasoning or practical scenarios that justify this design choice?

**Ethical Concerns:**

["NO or VERY MINOR ethics concerns only"]

**Final Justification:**

The authors have thoroughly addressed my concerns and questions with clear explanations and well-substantiated reasoning. Notably, the study directly tackles the dual challenges of station heterogeneity and oversmoothing, and demonstrates meaningful mitigation in both aspects. Their responses consistently demonstrate the depth and rigor of the work. Taken together, I maintain my support for accepting this paper.

**Limitations:**

Yes

**Paper Formatting Concerns:**

NA.

**Quality:**

3

**Strengths And Weaknesses:**

### Strengths

1.This work addresses a critical and previously underexplored problem in modern weather forecasting: transforming coarse-grained grid outputs into precise, station-level predictions. As the field moves toward high-resolution, localized, and user-relevant forecasts, this contribution is not only timely but strategically essential.

2.While individual model components are grounded in prior work, the authors demonstrate clear and deliberate architectural innovation by tailoring these modules to the unique challenges of grid-to-station correction. This task-driven refinement is thoughtful and well-justified, representing a meaningful form of innovation.

3.The paper is clearly written and well-organized, with figures and narrative consistently aligned around the two central challenges. The analysis is easy to follow, and the results provide concrete evidence that both identified issues are effectively addressed.

4.The experimental evaluation is thorough, with comparisons against diverse baselines across multiple forecast sources, variable transfer tests, and sensitivity analyses. The release of code and training details further supports reproducibility and transparency.

### Weaknesses

1. The paper does not report the parameter size or computational cost of the proposed method compared to baselines, which would help assess its efficiency and practicality.

2. Some parts of the method are described with limited detail, which may affect clarity and reproducibility for readers less familiar with the setting.

---

> ### Author Rebuttal · Authors · 2025-07-30
>
> Thank you so much for recognizing the value of our work and for your willingness to support its acceptance! We truly appreciate your encouraging feedback and it means a lot to us! To help you better understand our work, we’ve addressed the questions you raised in more detail below.
>
> > Q1:Can the authors provide a more explicit evaluation of the model's performance across different seasons, based on the seasonal precipitation error analysis shown in the appendix?
>
> Thank you for this helpful question! In response, we have **conducted an explicit evaluation of our model's performance across different seasons**, following the seasonal precipitation partitioning shown in the appendix. We compared our method with linear interpolation under the same seasonal split and summarized the results in the table provided.
>
> As the results show, our method consistently outperforms linear interpolation across all seasons in terms of RMSE, accuracy, and threshold-based scores. Notably, during summer when precipitation is most intense and variable, **our model demonstrates a clear improvement in capturing high-magnitude events.** This suggests that it effectively mitigates the over-smoothing issue commonly seen in traditional methods and achieves significantly enhanced predictive performance under challenging conditions.
>
> **Seasonal Evaluation Results**
> | Season  | Method     | RMSE   | ACC    | TS@2mm | TS@20mm | TS@100mm |
> |---------|------------|--------|--------|--------|----------|----------|
> | Spring  | DiffLiG    | 5.3105 | 0.7354 | 0.474  | 0.3618   | 0.02281   |
> |         | Linear     | 6.4889 | 0.6364 | 0.4403 | 0.3191   | 0.02113   |
> | Summer  | DiffLiG    | 9.1056 | 0.6108 | 0.467  | 0.3695   | 0.02749   |
> |         | Linear     |10.2558 | 0.5337 | 0.4126 | 0.3214   | 0.02386   |
> | Autumn  | DiffLiG    | 3.6167 | 0.7845 | 0.4531 | 0.2951   | 0.01760   |
> |         | Linear     | 4.4120 | 0.6857 | 0.3760 | 0.2531   | 0.01244   |
> | Winter  | DiffLiG    | 1.5181 | 0.8931 | 0.4593 | 0.2328   | 0.01339   |
> |         | Linear     | 2.0364 | 0.8178 | 0.3964 | 0.1918   | 0.00887   |
>
> > Q2:Can the authors provide a comparison of the parameter counts or model complexity between the proposed method and the baselines?
>
> To provide a fair and transparent comparison of model complexity, we evaluated all methods using a unified pipeline that reports the number of trainable parameters, model size, per-sample FLOPs, peak GPU memory usage, and average inference time. These results demonstrate that our method achieves significantly improved performance while introducing only a modest and controllable increase in model complexity. **The detailed comparison of model size, training cost, and inference efficiency is provided in Table 1.** We also re-evaluated the baseline methods through multiple independent runs to ensure fairness and stability. **The results are reported in Table 2.**
>
> > Q3:Could the authors clarify how this method relates to existing super-resolution approaches? If higher-resolution inputs become available, would this framework still provide complementary benefits at the station level?
>
> While our method shares the goal of improving spatial detail with super-resolution techniques, **it follows a fundamentally different path**. Traditional super-resolution requires upsampling the entire forecast field, which is computationally demanding. As highlighted in *The Quiet Revolution of Numerical Weather Prediction* (Bauer et al., 2015), increasing spatial resolution often leads to exponential growth in computational cost due to finer grids and smaller time steps. Even AI-based models, though more efficient than traditional NWP, still face a steep rise in training and inference costs as resolution increases. What's more, AI-based methods are typically trained on reanalysis datasets, which are themselves generated by physical models.
>
> In contrast, **our method is lightweight and targeted.** It directly interpolates gridded forecasts to station locations using a graph-based approach, allowing training on a single NVIDIA 4090D GPU within 6–7 hours. It avoids the need for global field reconstruction while delivering fine-scale accuracy where it matters most.
>
> Importantly, our framework is resolution-agnostic and fully compatible with future high-resolution forecasts. As upstream models improve, **our method can naturally leverage their outputs to further enhance station-level performance** without requiring any changes to their internal structure.
>
> > Q4:In the variable transfer experiment from precipitation to temperature, what components or hyperparameters of the model were adjusted to achieve the reported results?
>
> As noted in line 662 of the manuscript, *"the network structure remains unchanged."* To assess the robustness of our model across different meteorological variables, **we deliberately kept all model settings and hyperparameters identical** when transferring from precipitation to temperature. This design choice was intended to validate the generalizability of our framework without relying on task-specific tuning.
>
> In the context of weather prediction, **precipitation is widely regarded as one of the most challenging variables to forecast** due to its high variability and spatiotemporal discontinuity. In contrast, temperature tends to be more stable and easier to predict. This is also why we focus primarily on precipitation in the main experiments.
>
> > Q5:What was the motivation behind including future station observations at time T2 in the problem formulation? Could the authors elaborate on the reasoning or practical scenarios that justify this design choice?
>
> The inclusion of future time steps (denoted as T2) in our problem formulation is **motivated by a common practical scenario in operational forecasting.** In many real-world applications, we aim to estimate station-level values for future days, for which ground-truth observations are not yet available. However, model-generated forecasts at these future times are already accessible.
>
> Our framework is designed to bridge this gap by interpolating from the gridded model forecasts at time T2 to predict station-level values at the same future time step. This setup reflects a realistic and meaningful use case: making station-level predictions ahead of time using available gridded forecasts, even when the corresponding station observations do not yet exist.
>
> > Q6:Some parts of the method are described with limited detail.
>
> **Due to space limitations** in the main text, we were not able to include all implementation details. Instead, we focused on presenting the key components and novel contributions of our method in a concise manner.
>
> **To ensure clarity and reproducibility, we have released the full codebase.** We believe this provides readers with the necessary resources to fully understand and replicate our approach. We sincerely appreciate your understanding and are open to including more technical details in an extended version or supplementary materials if needed.
>
> > **Table 1: Model Complexity Comparison Table**
>
> | Model         | P (#)   | S (MB) | F (M)   | M (MB) | T (s) |
> |---------------|---------|--------|---------|--------|--------|
> | DiffLiG (ours)| 49,361  | 0.19   | 2383.73 | 252.5  | 161    |
> | LiGAP         | 39,397  | 0.15   | 1737.22 | 212.44 | 154    |
> | GAP           | 33,650  | 0.13   | 1534.93 | 189.76 | 144    |
> | Graph-bias    | 29,506  | 0.11   | 1462.59 | 182.52 | 143    |
> | Graph         | 24,930  | 0.10   | 1462.58 | 182.50 | 143    |
> | GCN           | 29,031  | 0.11   | 1233.01 | 166.07 | 137    |
> | GAT           | 29,287  | 0.11   | 1253.18 | 226.30 | 150    |
> | GraphSAGE     | 33,127  | 0.13   | 1832.93 | 186.42 | 146    |
> | KCN           | 30,520  | 0.12   | 1309.42 | 171.80 | 141    |
> | ViT           | 51,538  | 0.20   | 3972.84 | 2588.59| 326    |
> | Diffusion     | 54,014  | 0.21   | 2340.91 | 3793.07| 280    |
> | MeshGraphNet  | 41,863  | 0.16   | 2234.15 | 223.91 | 156    |
> | MGNN          | 40,741  | 0.16   | 2464.29 | 224.25 | 158    |
>
> **Note:**
> *P: Number of Parameters | S: Model Size | F: Per-sample FLOPs | M: Peak GPU Memory Usage | T: Average Inference Time per Epoch*
>
> We conduct ablation experiments to analyze the contribution of each module. LiGAP removes the diffusion module from our full model. GAP further removes the GeoLiquidNet and temporal attention. Graph-bias disables the dynamic edge modulation, and Graph additionally removes the station bias term.
>
> > **Table 2: Performance Comparison of Baselines**
>
> | Model         | RMSE      | ACC      | TS@2mm     | TS@20mm    | TS@100mm   |
> |---------------|----------------|---------------|------------------|------------------|------------------|
> | DiffLiG (ours)| 4.9971 ± 0.102 | 0.7411 ± 0.012 | 0.4637 ± 0.005   | 0.3122 ± 0.001   | 0.02019 ± 0.001  |
> | LiGAP         | 5.1589 ± 0.096 | 0.7321 ± 0.013 | 0.4559 ± 0.003   | 0.2819 ± 0.001   | 0.01473 ± 0.001  |
> | Linear        | 5.9033         | 0.6667         | 0.4011           | 0.2764           | 0.01558          |
> | GAT           | 5.4984 ± 0.095 | 0.6667 ± 0.015 | 0.3202 ± 0.012   | 0.2986 ± 0.001   | 0.0162 ± 0.001   |
> | GCN           | 5.4832 ± 0.055 | 0.6947 ± 0.012 | 0.4139 ± 0.011   | 0.2836 ± 0.002   | 0.01314 ± 0.001  |
> | GraphSAGE     | 5.6421 ± 0.003 | 0.6495 ± 0.009 | 0.3754 ± 0.003   | 0.2648 ± 0.002   | 0.01571 ± 0.001  |
> | KCN           | 5.3897 ± 0.008 | 0.6932 ± 0.012 | 0.4158 ± 0.007   | 0.2862 ± 0.002   | 0.01357 ± 0.002  |
> | MeshGraphNet  | 5.5322 ± 0.011 | 0.6767 ± 0.010 | 0.4374 ± 0.005   | 0.2925 ± 0.001   | 0.01443 ± 0.001  |
> | MGNN          | 5.4428 ± 0.116 | 0.6834 ± 0.013 | 0.3640 ± 0.009   | 0.2611 ± 0.001   | 0.01531 ± 0.001  |
>
> **We sincerely appreciate your recognition of our work! and your further endorsement would be a great encouragement to us.**

---

> > ### Comment · Reviewer_nrpT · 2025-08-04
> >
> > The authors have thoroughly addressed my concerns and questions with clear explanations and well-substantiated reasoning. Notably, the study directly tackles the dual challenges of station heterogeneity and oversmoothing, and demonstrates meaningful mitigation in both aspects. Furthermore, the additional analysis on computational complexity shows that the method remains lightweight and practical compared to baseline approaches. Moreover, I observed that in response to other reviewers' critiques, the authors provided comprehensive and thoughtful replies, supported by additional experiments and solid justifications. Their responses consistently demonstrate the depth and rigor of the work. Taken together, I believe this is a meaningful and well-executed contribution that opens up a promising direction for tackling practical challenges in meteorological forecasting. I strongly recommend the acceptance of this paper.

---

> > > ### Author Response · Authors · 2025-08-04
> > >
> > > We sincerely thank you for your generous recognition and thoughtful support! Your thoughts resonate deeply with ours. Our intention from the beginning has been to explore a distinct and meaningful path toward addressing the challenges of meteorological forecasting. If there is anything more you would like to discuss with us, we would be glad to continue the conversation. Once again, we truly appreciate your constructive feedback and the time you have dedicated to engaging with our research.

---

### Official Review · Reviewer_hRjM · 2025-07-01

**Clarity:** 2
**Significance:** 3
**Originality:** 3
**Rating:** 4
**Confidence:** 3

**Summary:**

This paper proposes a novel approach for Precipitation Correction based on graph learning. In details, to avoid oversmoothing and fully leverage the rich information provided by the station-level heterogeneity, the authors introduce a GeoLiquidNet which adapts temporal encoding and a GNN to dynamicly learn spatially adaptive connectivity. Additionally, a probabilistic diffusion selector for ensemble forecasts generation is designed to avoid oversmoothing.

**Questions:**

My main concenrs are as follows:
1. Compared to DDPM, the novelty of GeoLiquidNet seems limited. And some designs are not well explained.
2. The clarity of the figures.
3. How the oversmoothing problem is sovled.

**Ethical Concerns:**

["NO or VERY MINOR ethics concerns only"]

**Final Justification:**

The authors' rebuttal has addressed all my concerns.

However, due to the figures in the paper need to be revised for better clarity and some mathematical symbols require correction, I'm not entirely confident that the revised version will be significantly easier for readers to understand. In addition, the explanation of over-smoothing is particularly important for understanding the proposed method and the discussion of the problem. I believe the authors should provide more elaboration on this aspect in the main text rather than placing it in the appendix.

Therefore, taking all these factors into account, along with my confidence level in this area, I have decided to raise my rating to 4 instead of 5.

**Limitations:**

Yes

**Quality:**

3

**Strengths And Weaknesses:**

Pros:
1. This paper focuses on meteorological and remote sensing problems, which represent a real-world scenario with significant research value.
2. The authors conduct extensive experiments to prove the effectiveness of the proposed method.

Cons:
1. Although facing different scenarios, the GeoLiquidNet is essentially a DDPM-like model, merely applied to the meteorological domain to simulate historical evolution. At its core, it still relies on Gaussian noise for perturbation and generative modeling. Why does employing a Markovian-style diffusion process, along with location-conditioned time-step control, allow the model to flexibly adapt to diverse temporal rhythms across stations? Where exactly does the ``flexibility'' of such a Markov process come from? And what is the design rationale behind this time-step control mechanism?
2. Some symbols are not explained. For example, what do $W_1$ and $W_2$ represent. And some symbols are used more than once. For example, in line 159, $\kappa$ denotes the trainable dynamic coefficients, however, in line 175, $\kappa$ denotes the decay factor. The authors should re-check the whole paper for such errors.
3. The Figure 2 uses an excessive number of fonts and colors, making it difficult to understand the authors' idea. Moreover, some text elements are hard to read due to poor font and color choices. The figure also lacks a legend, and several elements with different meanings are presented using similar graphical forms, which further adds to the confusion.
4. Why generating more samples could benefit for the oversmoothing problem? Wouldn't a more complex GNN further exacerbate the oversmoothing problem?
5. In Figure 1, the explanation of Challenge 2 is quite vague, making it difficult to understand how it demonstrates the over-smoothing problem.

---

> ### Author Rebuttal · Authors · 2025-07-30
>
> We sincerely thank you for taking the time to engage with our work. We hope that our following clarifications will help address the concerns raised and offer a clearer understanding of our design motivations.
>
> > GeoLiquidNet Module
>
> While we understand the comparison to DDPM-like models, we would like to clarify that **our method does not follow the generative diffusion paradigm.** The Gaussian noise term in Eq. (1) serves only as a light perturbation to enhance robustness and temporal variability modeling, rather than playing a central role in the learning or prediction process.
>
> Our model is built upon a continuous-time OU process, where the hidden state evolves according to a location-dependent update rule. **The key to its flexibility lies in a time-step controller** that adapts the integration speed based on each station’s geographic characteristics. Stations with rapid precipitation changes receive shorter time constants for quick adaptation, while stable regions use longer memory for smoothing.
>
> In essence, this mechanism functions as a learnable, location-aware low-pass filter. We provide a detailed theoretical explanation of this behavior in **Appendix E**.
>
> To help clarify the distinction between our approach and DDPM-style designs, we conducted **a set of ablation studies** where we replaced GeoLiquidNet’s temporal module with several alternatives, including a DDPM-style embedding, a standard MLP-based embedding, and an attention-based embedding.
> | Method        | RMSE             | ACC         | TS@2mm        | TS@20mm        | TS@100mm        |
> |---------------|------------------|------------------|------------------|------------------|------------------|
> | DiffLiG (ours) | 4.9971 ± 0.102   | 0.7411 ± 0.012   | 0.4637 ± 0.005   | 0.3122 ± 0.001   | 0.02019 ± 0.001   |
> | DiffDdpmG      | 5.1230 ± 0.071   | 0.7321 ± 0.013   | 0.4521 ± 0.003   | 0.2894 ± 0.002   | 0.01908 ± 0.001   |
> | DiffMlpG       | 5.1011 ± 0.061   | 0.7351 ± 0.006   | 0.4441 ± 0.003   | 0.2911 ± 0.002   | 0.01801 ± 0.001   |
> | DiffAttG       | 5.1284 ± 0.023   | 0.7388 ± 0.011   | 0.4437 ± 0.005   | 0.2933 ± 0.002   | 0.01833 ± 0.001   |
>
> > Symbol Usage
>
> We sincerely apologize for the confusion and inconvenience this may have caused you, and we truly appreciate your careful reading and constructive feedback.
>
> We apologize for the confusion caused by the ambiguous or duplicated use of certain symbols in the manuscript. Specifically, **W₁** and **W₂** refer to two independent linear projection layers implemented in our codebase (DiffLiG/Source/Modules/GNN/liquidnet.py , lines 143 and 162).
>
> Regarding the repeated use of the same symbol for different meanings, we initially intended to distinguish them through typesetting: the former was bolded to indicate vector-valued parameters, while the latter included a subscript to denote a scalar variable. We will revise these symbols thoroughly in the next version to eliminate ambiguity and ensure consistency throughout the manuscript.
>
> > Figure 1
>
>  **Figure 1 is not intended to illustrate the underlying mechanism of the over-smoothing problem in detail.** Rather, it serves as a comic-style overview to highlight the two main challenges addressed in our work: station heterogeneity and over-smoothing.
>
> Specifically for Challenge 2, the upper solid raindrops represent predicted precipitation, while the lower dashed raindrops indicate missed or smoothed-out extremes. The illustration of the Forecast Diffuser & Selector below aims to convey that, in reality, higher rainfall should be retained, whereas under-predicted low rainfall may reflect the failure to capture extremes due to over-smoothing. This visual cue is symbolic rather than analytical, and is not meant to convey precise technical details.
>
> For a rigorous and systematic discussion, we refer you to Appendix D, where we provide detailed experiments and quantitative analysis.
>
> > Figure 2
>
> The use of multiple colors and font styles in Figure 2 was originally intended to visually separate different modules in our framework, given its relatively complex structure.
>
> To briefly clarify the structure shown in Figure 2:
>
> - The upper half of the figure illustrates the full model pipeline, starting from the station-level inputs (left), going through the GeoLiquidNet and Dynamic Edge Modulator, leading to the Probabilistic Forecast Diffuser and Multi-Criteria Score Aggregator (right).
> - The lower half expands on the internal designs of two core components: the Dynamic Edge Modulator (bottom left) and the Probabilistic Forecast Diffuser (bottom right).
>
> The figure is meant to align closely with the corresponding modules and subsection names in the main text, and their **ordering is kept consistent between the figure and the written descriptions.**
>
> We truly value your suggestion. In the revised version, we will work to simplify the visual layout, unify the typography, and add a legend to improve readability.
>
> > Origin and Mitigation of the Over-Smoothing Problem
>
> **Sources of oversmoothing.** Oversmoothing in our task stems mainly from two sources: the inherent bias in ERA5 data, which tends to smooth out local extremes (see Appendix D), and the model’s tendency to underestimate heavy precipitation in pursuit of better global metrics. To evaluate this, we introduced TS@20mm and TS@100mm as targeted indicators, and found that all models exhibit low scores under these thresholds, confirming the difficulty of accurately capturing such extremes.
>
> **Diffusion design as the core solution.** Thank you for the thoughtful question. We would like to clarify that the over-smoothing problem in our model is not addressed simply by generating multiple samples. Instead, the core of our solution lies in introducing a diffusion-based mechanism. As we discuss in Appendix F, this design allows the model to move beyond average-case behavior and better capture rare but significant precipitation events. The purpose of generating multiple samples is to enable probabilistic forecasting and to improve the prediction accuracy through ensemble aggregation. To further verify this, we include additional experiments varying the number of generated samples. Notably, even when only one sample is generated, the model still shows clear improvement in capturing heavy rainfall.
> | Samples |   RMSE    | ACC     | TS@2mm    | TS@20mm  | TS@100mm  |
> |-----------|----------------|---------------|----------------|----------------|----------------|
> | 1         | 5.1241 ± 0.101 | 0.7332 ± 0.010 | 0.4611 ± 0.004 | 0.2992 ± 0.001 | 0.01839 ± 0.001 |
> | 3         | 5.0654 ± 0.099 | 0.7386 ± 0.015 | 0.4622 ± 0.003 | 0.3012 ± 0.001 | 0.02016 ± 0.001 |
> | 5         | 4.9971 ± 0.102 | 0.7411 ± 0.012 | 0.4637 ± 0.005 | 0.3122 ± 0.001 | 0.02019 ± 0.001 |
> | 7         | 4.9995 ± 0.131 | 0.7501 ± 0.016 | 0.4651 ± 0.003 | 0.3132 ± 0.002 | 0.01996 ± 0.001 |
> | LiGAP     | 5.1589 ± 0.096 | 0.7321 ± 0.013 | 0.4559 ± 0.003 | 0.2819 ± 0.001 | 0.01473 ± 0.001 |
>
> **Model capacity is not the solution.** To further examine whether the over-smoothing issue could be attributed to model capacity, we constructed a Large_LiGAP variant by significantly increasing the number of parameters in LiGAP. As shown in the table below, this larger model achieves slightly better RMSE and ACC, but it fails to improve TS@20mm and TS@100mm. In fact, its performance on extreme events remains nearly identical to the original LiGAP. However, we do observe that larger models tend to converge in fewer training epochs.
>
> | Model        | RMSE      | ACC      | TS@2mm     | TS@20mm    | TS@100mm   |
> |--------------|----------------|----------------|------------------|------------------|------------------|
> | Large_LiGAP  | 5.1289 ± 0.011  | 0.7371 ± 0.011  | 0.4607 ± 0.009   | 0.2815 ± 0.001   | 0.01470 ± 0.001  |
>
> These results demonstrate that simply increasing the model capacity does not alleviate the over-smoothing problem. In contrast, DiffLiG significantly improves the TS metrics at high thresholds, confirming that our diffusion-based design plays a key role in mitigating over-smoothing. The detailed parameter comparison can be found in *Table 1*.
>
>
> **Two-fold evaluation of oversmoothing mitigation.** The mitigation of over-smoothing can be evaluated from two complementary perspectives. First, we selected a representative day with widespread extreme precipitation and visualized the station-level predictions. As shown in Appendix N, our method achieves better spatial fidelity in recovering the intensity and variability of heavy rainfall. Second, we added two additional metrics, TS@20mm and TS@100mm. The improvements on both metrics suggest that our approach is effective in alleviating over-smoothing and enhancing the model's capacity to express extreme values.
>
> **We have organized the performance metrics and computational efficiency of the re-evaluated baselines into tables. Due to space limitations, these results are included in the response to Reviewer nrpT, specifically in Table 1 and Table 2.**
>
> > Table 1: Model Complexity Comparison Table
>
> | Model         | P (#)   | S (MB) | F (M)   | M (MB) | T (s) |
> |---------------|---------|--------|---------|--------|--------|
> | DiffLiG (ours)| 49,361  | 0.19   | 2383.73 | 252.5  | 161    |
> | DiffLiG1      | 49,361  | 0.19   | 2229.74 | 611.67 | 159    |
> | DiffLiG3      | 49,361  | 0.19   | 2306.73 | 611.67 | 159    |
> | DiffLiG7      | 49,361  | 0.19   | 2460.71 | 611.67 | 161    |
> | Large_LiGAP   | 369,858 | 1.41   | 22627.45| 2604.42| 221    |
> | DiffDdpmG     | 53,047  | 0.20   | 2201.16 | 212.35 | 153    |
> | DiffMlpG      | 42,163  | 0.16   | 1747.09 | 189.80 | 158    |
> | DiffAttG      | 49,811  | 0.19   | 1979.48 | 215.96 | 173    |
>
> **Note:**
> *P: Number of Parameters | S: Model Size | F: Per-sample FLOPs | M: Peak GPU Memory Usage | T: Average Inference Time per Epoch*
>
> **We would be glad to further discuss this with you, as your feedback is truly valuable to us.**

---

> > ### Comment · Reviewer_hRjM · 2025-08-05
> >
> > Thanks for the authors' response. I think they have addressed all my concerns. I'm willing to raise my score.
> >
> > However, due to the figures in the paper need to be revised for better clarity and some mathematical symbols require correction, I'm not entirely confident that the revised version will be significantly easier for readers to understand. In addition, the explanation of over-smoothing is particularly important for understanding the proposed method and the discussion of the problem. I believe the authors should provide more elaboration on this aspect in the main text rather than placing it in the appendix.
> >
> > Therefore, taking all these factors into account, along with my confidence level in this area, I have decided to raise my rating to 4 instead of 5.

---

> > > ### Author Response · Authors · 2025-08-05
> > >
> > > Thank you very much for your feedback! We sincerely appreciate your recognition of our work and your willingness to raise the rating. Your encouragement means a great deal to us. We also value your thoughtful and constructive suggestions, which have helped us further improve the clarity and completeness of the paper.
> > >
> > > In the revised version, we will carefully update the figures to improve their visual clarity and correct the mathematical symbols as suggested. We fully agree that the explanation of over-smoothing is essential for understanding our method, and we will therefore move this discussion into the main text and provide additional elaboration.
> > >
> > > Once again, thank you for taking the time to engage with our work. Your feedback has played an important role in guiding our revisions and strengthening the manuscript. If you have any further questions, we would be glad to continue the discussion.

---

> ### Author Response · Authors · 2025-08-04
>
> Dear Reviewer hRjM,
>
> Thank you again for your thoughtful and constructive feedback. We sincerely hope that the revisions and clarifications we have provided have resolved the concerns you raised.
>
> If there remain any outstanding questions or points requiring clarification, we would be more than happy to elaborate further. Otherwise, if you feel that our responses have addressed your concerns satisfactorily, **we would be most grateful if you could consider raising your score.**
>
> We truly appreciate your valuable time and engagement, and **we look forward to any further comments you may wish to share.**
>
> Best regards,
>
> The Authors

---

### Official Review · Reviewer_cW4J · 2025-07-02

**Clarity:** 2
**Significance:** 1
**Originality:** 1
**Rating:** 3
**Confidence:** 4

**Summary:**

This paper proposes DiffLiG, a complex three-component graph neural network for correcting gridded meteorological forecasts to station-level observations. The architecture combines GeoLiquidNet (using Ornstein-Uhlenbeck dynamics), Dynamic Edge Modulator, and Probabilistic Diffusion Selector. While evaluated on Chinese meteorological data, the paper fails to justify why such complexity is needed for what is essentially a spatial interpolation task.

**Questions:**

Why is graph-based modeling necessary at all? Your ablation study keeps the GCN in all variants. What happens if you remove ALL graph components and use a simple MLP that directly maps from grid points to stations? This fundamental comparison is missing.
Why use GCN when recent work shows transformer-based architectures are superior for spatial tasks? Vision Transformers and spatial attention mechanisms have shown better performance on geographic data. Why stick with message-passing GNNs that are known to suffer from oversmoothing?
Why not use Neural Operators for this continuous field interpolation? Recent work on Fourier Neural Operators and DeepONet are specifically designed for function approximation between continuous fields. Your grid-to-station task seems perfectly suited for these architectures - why force it into a graph framework?
Have you considered Implicit Neural Representations (INRs)? Methods like SIREN or Neural Fields can represent continuous spatial functions and naturally interpolate to any coordinate. This would eliminate your complex graph construction entirely. Why wasn't this explored?
Why use outdated GCN when Graph Transformers exist? Recent architectures like GraphGPS, SAN, or Graphormer address many limitations of GCNs. Your 2-layer GCN seems particularly primitive given advances in graph architectures.
Why separate temporal and spatial modeling? Recent spatiotemporal architectures like STGNNs, DCRNN, or Space-Time Transformers jointly model both dimensions. Your sequential approach (temporal then spatial) seems suboptimal.
Why not use Conditional Normalizing Flows for uncertainty? Your diffusion-based ensemble seems overcomplicated compared to flow-based models that can directly model the conditional distribution P(station|grid) with exact likelihood.
Have you tried Cross-Attention mechanisms? Instead of fixed graph edges, cross-attention between grid and station features would be more flexible and learnable. Why use hard-coded k-nearest neighbors?
What is the computational cost of your three-module architecture? Please provide concrete runtime and memory comparisons with baselines. How can practitioners justify such complexity for a spatial interpolation task?
Where is the statistical validation? Your improvements are often less than 5%. Without error bars, confidence intervals, or significance tests, how do we know these aren't within measurement noise? Please provide proper statistical analysis.
Why don't you evaluate on extreme events? Your introduction emphasizes extreme precipitation, yet your evaluation uses standard metrics on all data. What is the performance specifically on the top 1% or 5% of rainfall events?
How does your method perform with limited historical data? You assume 5 days of historical observations, but many stations have gaps. What happens with only 1-2 days of history, or none at all?

**Ethical Concerns:**

["NO or VERY MINOR ethics concerns only"]

**Final Justification:**

After careful consideration of the authors' rebuttal and additional experiments, I am raising my score from 2 to 3.

Issues that were resolved:

The authors provided comprehensive ablation studies comparing with MLP, Transformer, SIREN, and other architectures, addressing my concern about whether graph-based modeling is necessary
Statistical validation with error bars and significance tests was added, addressing the lack of statistical rigor
Extreme event evaluation metrics (TS@20mm, TS@100mm) were included, partially addressing the disconnect between motivation and evaluation
Computational cost analysis and cross-dataset validation (MADIS) demonstrate practical feasibility
The clarification that linear interpolation remains standard practice in meteorological agencies helps justify the practical value

Issues that remain partially unresolved:

The architecture still appears overly complex for a spatial interpolation task, though the authors argue it's computationally comparable to baselines
The biological inspiration (OU dynamics) for meteorological modeling remains scientifically questionable, despite the provided theoretical justification
Cross-regional generalization beyond China and US datasets would strengthen claims of general applicability

Weight of considerations:

The authors have made substantial efforts to address the core experimental weaknesses. I maintain some skepticism about the architectural complexity. The practical grounding from meteorological agencies and the computational efficiency make this work potentially adoptable.

**Limitations:**

The authors mention some limitations but miss critical ones:

No analysis of when the method fails
No computational cost discussion
No consideration of why operational agencies would adopt this over proven methods
The geographic limitation to China is downplayed

More concerning is what the paper doesn't acknowledge: this appears to be a solution in search of a problem.

**Quality:**

1

**Strengths And Weaknesses:**

Strengths:

The paper addresses station-level forecast correction, which has practical applications
Includes implementation details in appendices
Attempts to combine multiple deep learning techniques

Weaknesses:

Fundamentally flawed experimental design: The ablation study never removes the GCN backbone, making it impossible to determine if graph-based modeling is even beneficial. This is a fatal flaw that undermines the entire contribution
Unjustified architectural choices: The use of OU dynamics from "biological inspiration" for meteorological data is scientifically questionable. No evidence provided that this offers any advantage over standard approaches
Missing essential baselines: No comparison with simple MLPs (to justify graphs), AGCRN (state-of-the-art adaptive GNN), or operational weather post-processing systems
No statistical validation: Despite marginal improvements (~5%), no error bars, confidence intervals, or significance tests provided


Overly complex presentation obscures lack of innovation
Core claims unsupported: "heterogeneity" and "oversmoothing" presented as major challenges without evidence (other factors may also be challenging)
Contradictory motivations: emphasizes extreme events but never evaluates on them


Evaluated only on China, only on precipitation (temperature relegated to appendix)
No cross-regional validation, no extreme weather evaluation
Improvements too marginal to justify adoption
No theoretical insights, purely incremental engineering


All components borrowed from existing work without principled integration
Appears to chase trends (diffusion models) rather than solve problems effectively
No novel contributions to understanding spatial interpolation or meteorological processes

---

> ### Author Rebuttal · Authors · 2025-07-30
>
> We sincerely thank you for the thoughtful comments. While some concerns were raised about the value of our work, we would like to clarify that our study is in fact grounded in practical needs and solid experimental design.
>
> > Significance of This Work
>
> Our work is motivated by practical forecasting applications that require accurate, station-level precipitation estimates, such as flash flood alerts, and localized hydrological planning. However, **current large-scale models typically produce gridded forecasts**, which do not directly provide rainfall values at station locations.
>
> **Generalization beyond precipitation.** Our method is proposed as a general framework to bridge this gap. Although we primarily evaluate it on precipitation due to its high variability and modeling difficulty, the framework itself is broadly applicable. As demonstrated in Appendix G, our model transfers well to temperature. Due to space limitations, we focus on precipitation in the main text. We have included additional experiments using the MADIS dataset from the United States and evaluated the wind vector variable, as detailed in our response to Reviewer s17Q. You are welcome to refer to that section if interested.
>
> **ERA5 and station data mismatch.** Many forecasting studies use ERA5 as a reference, but it differs significantly from station observations, as shown in *Evaluation of spatial–temporal variation performance of ERA5 precipitation data in China* (Scientific Reports, 2021). We further demonstrate this discrepancy through experiments in Appendix C. As a result, meteorological agencies typically prefer to evaluate forecasts directly against station-level observations.
>
> **Linear interpolation as the mainstream practice.** We consulted experts from the Meteorological Administration, who confirmed that linear interpolation remains the most commonly used method for post-processing gridded forecasts. We include comparisons with this baseline in our paper.
>
> > station heterogeneity and oversmoothing
>
> As analyzed in Appendix D, station heterogeneity stems from applying uniform interpolation across sites with differing climate conditions, while oversmoothing arises as models tend to predict averaged values to minimize global error.
>
> > GNN as Framework
>
> At the beginning, we agree that spatial interpolation tasks ideally call for lightweight and simple solutions. In fact, our model is designed with efficiency in mind: **it can be trained on a single 4090D GPU**, and its parameter count and training time are comparable to baseline methods. Our goal from the outset has been to develop a compact model that still delivers superior performance.
>
> In the early phase of this project, **we explored a range of architectural choices.** We noticed that the MGNN paper had already experimented with pure MLP-based interpolation. For this reason, we only included MGNN in our main comparisons. Beyond that, we also considered FNO and Transformer models. A major challenge with these approaches is determining how many grid points to include for each station. When adopting global modeling, the dense grid and sparse station setup led to vanishing gradient issues during training.  In fact, observations at a station are typically influenced by a few nearby grid cells. This locality makes GNNs a natural fit for the task.
>
> **we also implemented corresponding experiments**. In practice, the GNN structure we adopt is not as complex as it may seem. It simply connects each station to a few nearby grid points. As shown by the parameter counts, we deliberately scaled up the alternative models to match or exceed our own model's size in order to ensure fair comparisons.
> | Model       | RMSE            | ACC             | TS@2mm           | TS@20mm         | TS@100mm         |
> |-------------|------------------|------------------|------------------|------------------|------------------|
> | Linear      | 5.9033           | 0.6667           | 0.4011           | 0.2764           | 0.01558           |
> | MLP         | 6.6446 ± 0.088   | 0.4324 ± 0.013   | 0.2975 ± 0.003   | 0.0723 ± 0.001   | 0.0095 ± 0.001   |
> | Transformer | 6.7028 ± 0.092   | 0.4122 ± 0.011   | 0.2713 ± 0.005   | 0.0728 ± 0.001   | 0.0136 ± 0.001   |
> | SIREN       | 6.3095 ± 0.072   | 0.4967 ± 0.011   | 0.2910 ± 0.003   | 0.0751 ± 0.001   | 0.0084 ± 0.001   |
>
> > Graph Design
>
> We would like to clarify that **our model is not a simple GCN**, but incorporates multiple structural enhancements. These include a Dynamic Edge Modulator that adaptively controls edge influence, and a multi-head spatial attention mechanism that models asymmetric spatial interactions. we have compared against AGCRN and Graphormer, as you rightly mentioned.
>
> **Sequential modeling with joint fusion.** As for spatiotemporal modeling, we apply temporal and spatial modules sequentially, but their outputs are fused through shared MLP layers to enable joint reasoning. This approach is consistent with many STGNNs, as summarized in *Zhang et al., 2022, Spatio-Temporal Graph Neural Networks for Predictive Learning in Urban Computing: A Survey*.
>
> We would like to clarify that **our graph edges are not fixed.** They are adaptively modulated through a learnable Dynamic Edge Modulator. In our early experiments, we also explored more flexible structures. However, this led to training instability and vanishing gradient issues, likely due to the sparse and noisy supervision inherent in this interpolation setting. Moreover, introducing fully learnable connectivity would increase the model’s complexity. By using k-nearest neighbor connections as the initial structure and refining them through edge modulation, we achieve a solution that is stable, interpretable, and adaptable for real-world deployment.
> | Model       | RMSE             | ACC              | TS@2mm           | TS@20mm         | TS@100mm         |
> |-------------|------------------|------------------|------------------|------------------|------------------|
> | Graphormer  | 5.3412 ± 0.1213  | 0.7123 ± 0.012   | 0.3872 ± 0.013   | 0.2911 ± 0.002   | 0.01628 ± 0.001   |
> | AGCRN       | 5.5676 ± 0.162   | 0.6587 ± 0.035   | 0.3927 ± 0.012   | 0.2745 ± 0.002   | 0.01423 ± 0.001   |
>
> > GeoLiquid Module
>
> The GeoLiquid module adapts the temporal memory based on station characteristics, acting as a learnable low-pass filter. We provide its theoretical foundation in **Appendix E**. To further evaluate its effectiveness, we replaced it with DDPM-style embedding, MLP, and attention-based variants for temporal embedding.
> | Method        | RMSE             | ACC         | TS@2mm        | TS@20mm        | TS@100mm        |
> |---------------|------------------|------------------|------------------|------------------|------------------|
> | DiffLiG (ours) | 4.9971 ± 0.102   | 0.7411 ± 0.012   | 0.4637 ± 0.005   | 0.3122 ± 0.001   | 0.02019 ± 0.001   |
> | DiffMlpG       | 5.1011 ± 0.061   | 0.7351 ± 0.006   | 0.4441 ± 0.003   | 0.2911 ± 0.002   | 0.01801 ± 0.001   |
> | DiffAttG       | 5.1284 ± 0.023   | 0.7388 ± 0.011   | 0.4437 ± 0.005   | 0.2933 ± 0.002   | 0.01833 ± 0.001   |
>
> > Historical Observation Data
>
> You asked how the model performs with limited historical data. We would like to point out that we already conducted a **detailed sensitivity analysis in Appendix H.2.** Additionally, we supplemented an experiment where no historical station data is available.
> | Setting     | RMSE         | ACC       | TS@2mm      | TS@20mm      |  TS@100mm     |
> |-------------|------------------|------------------|------------------|------------------|------------------|
> | No-History  | 5.1342 ± 0.103   | 0.7351 ± 0.011   | 0.4436 ± 0.004   | 0.2855 ± 0.001   | 0.01783 ± 0.001   |
>
> > Oversmoothing and Probabilistic Diffusion Selector
>
> Oversmoothing in our task stems mainly from two sources: **the inherent bias in ERA5 data**(see Appendix D), and **the model’s tendency to underestimate heavy precipitation** in pursuit of better global metrics. To evaluate this, we introduced TS@20mm and TS@100mm as targeted indicators for extreme rainfall, and found that all models exhibit low scores under these thresholds. We visualize a representative extreme rainfall event (Appendix N).
>
> We clarify that **our diffusion module is not as complex as it may seem.** To address your suggestion, we also implemented Conditional Normalizing Flows for uncertainty modeling.
> | Model         | RMSE             | ACC              | TS@2mm           | TS@20mm         | TS@100mm         |
> |---------------|------------------|------------------|------------------|------------------|------------------|
> | LiGAP         | 5.1589 ± 0.096   | 0.7321 ± 0.013   | 0.4559 ± 0.003   | 0.2819 ± 0.001   | 0.01473 ± 0.001   |
> | Flow-LiGAP    | 5.1965 ± 0.0113   | 0.7324 ± 0.021   | 0.4680 ± 0.004   | 0.2765 ± 0.001   | 0.01526 ± 0.001   |
>
> > Table 1: Model Complexity Comparison Table
>
> | Model         | P (#)   | S (MB) | F (M)   | M (MB) | T (s) |
> |---------------|---------|--------|---------|--------|--------|
> | DiffLiG (ours)| 49,361  | 0.19   | 2383.73 | 252.5  | 161    |
> | MLP           |176,257  | 0.67   | 3318.77 | 471.75 | 163    |
> | Transformer   | 84,801  | 0.32   | 3664.15 | 1367.81| 182    |
> | AGCRN         | 74,842  | 0.29   | 6446.77 | 1488.35| 179    |
> | Graphormer    | 70,725  | 0.27   |11170.28 | 4660.00| 182    |
> | DiffMlpG      | 42,163  | 0.16   | 1747.09 | 189.80 | 158    |
> | DiffAttG      | 49,811  | 0.19   | 1979.48 | 215.96 | 173    |
> | Flow-LiGAP    | 84,100  | 0.32   | 4450.80 | 419.50 | 173    |
>
> **Note:**
> *P: Number of Parameters | S: Model Size | F: Per-sample FLOPs | M: Peak GPU Memory Usage | T: Average Inference Time per Epoch*
>
> **Due to space limitations, more results are provided in our response to Reviewer nrpT, specifically in Table 1 and Table 2.**
>
> **We sincerely thank you for your valuable suggestions! Your final evaluation of our work is of great importance to us!**

---

> ### Author Response · Authors · 2025-08-04
>
> Dear Reviewer cW4J,
>
> Thank you again for your thoughtful and constructive feedback. We sincerely hope that the revisions and clarifications we have provided have resolved the concerns you raised.
>
> If there remain any outstanding questions or points requiring clarification, we would be more than happy to elaborate further. Otherwise, if you feel that our responses have addressed your concerns satisfactorily, **we would be most grateful if you could consider raising your score.**
>
> We truly appreciate your valuable time and engagement, and **we look forward to any further comments you may wish to share.**
>
> Best regards,
>
> The Authors

---

> > ### Comment · Reviewer_cW4J · 2025-08-05
> >
> > Thank you for your detailed rebuttal and the additional experiments you've conducted. I appreciate the effort you've put into addressing my concerns.
> > Your new experiments comparing with MLP, Transformer, SIREN, and other architectures help clarify why GNN is a reasonable choice for this task. The statistical validation with error bars and the extreme event metrics (TS@20mm, TS@100mm) also strengthen the evaluation. I'm particularly glad to see the computational cost analysis and cross-dataset validation on MADIS, which address important practical considerations.
> > While I still have some reservations about the overall complexity of the approach for what is fundamentally a spatial interpolation task, I acknowledge that you've made a good faith effort to justify your design choices and demonstrate practical value. The clarification about linear interpolation being the current standard practice in meteorological agencies also helps contextualize the contribution.
> > Based on your comprehensive responses and additional experiments, I am willing to raise my score. However, I encourage you to incorporate these clarifications and additional results into the main paper to make the contribution clearer to readers.

---

> > > ### Author Response · Authors · 2025-08-05
> > >
> > > We sincerely thank you for your recognition of our work and your willingness to raise the score! We are especially grateful for the many constructive and thoughtful comments you have provided throughout the review process. Your suggestions prompted us to conduct a series of additional experiments and clarifications, which we believe have significantly improved the quality and clarity of our work.
> > >
> > > As you recommended, we will incorporate these clarifications and results into the main paper to ensure our contributions are communicated more clearly.
> > >
> > > Once again, thank you for taking the time to engage so thoroughly with our work. If you have any further questions, we would be very happy to continue the discussion.

---

### Note · Authors · 2025-08-12

Dear AC and Reviewers,

We sincerely thank you for your thoughtful feedback, constructive suggestions, and valuable engagement throughout the review and discussion phases. We deeply appreciate the time and effort you have devoted to evaluating our work.

Following the rebuttal, we are pleased to observe that all four reviewers now express **a clear inclination to accept our paper.** During the rebuttal phase, we took every reviewer’s feedback seriously and conducted corresponding additional experiments to address their concerns. Through this process, reviewers have gained a deeper understanding of our work.

Overall, the rebuttal phase has further strengthened reviewers’ confidence in our work:

- Reviewer **nrpT** (*"I strongly recommend the acceptance of this paper."*) not only maintained but further **strengthened his already strong support** for our work after the rebuttal, reinforcing his view of its value and significance.

- Reviewer **s17Q** **maintained his initial inclination toward acceptance** after reading our responses, further confirming his recognition of our work.

- Reviewer **hRjM** (*"I think they have addressed all my concerns. I'm willing to raise my score."*) clearly stated that our reply had resolved his concerns and **gave a positive evaluation** of our work.

- Reviewer **cW4J** (*"Based on your comprehensive responses and additional experiments, I am willing to raise my score."*) highlighted that our thorough responses and the additional experiments provided during the discussion **successfully addressed his concerns.**

Taken together, **the reviews reflect a shared recognition of our work.** We sincerely hope that the final decision will reflect this consensus and that our work will be evaluated in a comprehensive and fair manner.

Thank you again for your time and consideration.

Best regards,
The Authors

---

### Decision · Program_Chairs · 2025-09-17

**Decision:**

Accept (poster)

**Comment:**

This paper presents a graph-based approach to improve station-level weather forecasting by correcting coarse-resolution forecasts derived from numerical and AI-based models. The method introduces three main contributions: (1) station-specific temporal modeling, (2) a spatially adaptive modulator, and (3) ensemble correction with uncertainty quantification.
To fully leverage the heterogeneity of station-level data and avoid oversmoothing, the authors propose GeoLiquidNet, which combines temporal encoding with a GNN that dynamically learns spatially adaptive connectivity. In addition, a probabilistic diffusion selector is designed to generate ensemble forecasts while mitigating oversmoothing. The authors have thoroughly addressed reviewer concerns by adding comprehensive ablation studies, providing statistical validation with error bars and significance tests, incorporating extreme-event metrics, including computational cost analysis and cross-dataset validation.